# From Loops to Oops: Fallback Behaviors of Language Models Under Uncertainty

## Abstract

Large language models (LLMs) often exhibit undesirable behaviors, such as hallucinations and sequence repetitions. We propose to view these behaviors as fallbacks that models exhibit under epistemic uncertainty, and investigate the connection between them. We categorize fallback behaviors — sequence repetitions, degenerate text, and hallucinations — and extensively analyze them in models from the same family that differ by the amount of pretraining tokens, parameter count, or the inclusion of instruction-following training. Our experiments reveal a clear and consistent ordering of fallback behaviors, across all these axes: the more advanced an LLM is (i.e., trained on more tokens, has more parameters, or instruction-tuned), its fallback behavior shifts from sequence repetitions, to degenerate text, and then to hallucinations. Moreover, the same ordering is observed during the generation of a single sequence, even for the best-performing models; as uncertainty increases, models shift from generating hallucinations to producing degenerate text and finally sequence repetitions. Lastly, we demonstrate that while common decoding techniques, such as random sampling, alleviate unwanted behaviors like sequence repetitions, they increase harder-to-detect hallucinations.

## 1 Introduction

While large language models (LLMs) have been known to generate human-like language remarkably well (Radford et al., 2019; Brown et al., 2020; Touvron et al., 2023; OpenAI et al., 2024), there are growing concerns about their propensity for undesirable behaviors, such as degenerate[1] or repetitive text (Holtzman et al., 2020), hallucinations (Ji et al., 2022; Zhang et al., 2023), and verbatim recollection of training samples when processing out-of-distribution inputs (Nasr et al., 2023a). Previous work (Kim et al., 2024; Snyder et al., 2023) has studied these phenomena, and suggested solutions, but has done so for each in isolation, without considering the interactions between them.

In this work, we propose that the undesired behaviors illustrated in Figure 1 can be viewed collectively as *fallback behaviors*, which emerge when the model faces *epistemic uncertainty*, namely uncertainty due to lack of knowledge (Hou et al., 2023). We aim to categorize and analyze the relationship between these behaviors across a range of LLMs. To this end, we create controlled yet natural settings, which force the generation of factual information while introducing varying levels of model uncertainty by construction. We then test three model families—`Pythia` (Biderman et al., 2023), `Llama 2` (Touvron et al., 2023), `Llama 3` (Meta AI, 2024), and `OLMo` (Groeneveld et al., 2024)—considering various factors that could influence the emergence of different fallbacks: (a) number of parameters, (b) number of pretraining tokens, (c) instruction-following training, and (d) decoding algorithms. We observe a clear ordering in the appearances of different fallbacks, as demonstrated in Figure 2 and persisting across all the above factors: repetitions are the simplest fallback, followed by degenerate text, and finally hallucinations as the most complex behavior.

We present evidence that increasing the strength of a model[2] tends to shift its fallback behavior to more complex forms, whereas weaker models rely on simpler behaviors like repetitive text. We demonstrate that the so-called strength of a model can be the result of increasing the model's parameter count, additional pretraining, or the addition of an instruction-following training phase.

---

[1]Degenerate text includes repetitive textual patterns and/or rephrasing of previously generated text.

[2]We consider prolonged training time, increased parameter count and inclusion of instruction tuned phase as "increasing models' strength" as these techniques are ubiquitous in creation of more advanced LLMs.

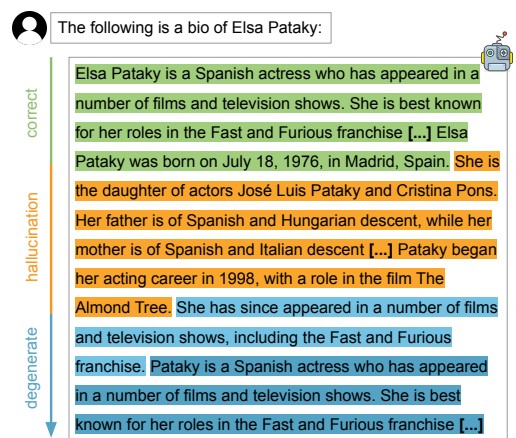

Figure 1: When language models face uncertainty, they exhibit fallback behaviors, shifting from hallucinations to degenerate text generation (repeating previous facts in different phrasing) and finally verbatim repetitions.

Moreover, we show that as generation length grows and the model struggles to generate factually correct responses or convincing hallucinations, it shifts back in the fallback behaviors spectrum from hallucinations to repetitions.

We further examine the effect of the decoding scheme, demonstrating that while random sampling may alleviate degenerate text (Holtzman et al., 2020), it increases the rate of hallucinations, which are harder to detect and potentially more damaging to the user. Despite evidence suggesting models may be partially aware of their knowledge gaps (Kadavath et al., 2022), they are not easily steered away from non-factual generations. Even when shown how to abstain rather than hallucinate, models continue to produce hallucinations.

Our study shows that although increasing model size reduces epistemic uncertainty (by enhancing parametric knowledge) and improves accuracy (Kaplan et al., 2020), when uncertainty arises, fallback behaviors persist even in the largest, best-trained models. Cases of uncertainty inevitably occur in natural interactions with LLMs, due to knowledge cutoffs, false premises in prompts, or questions about factual information not present in the training data. Ad-hoc fixes, like decoding schemes, fail to address the core issue, often replacing easily detectable degenerate text with subtler and potentially harmful hallucinations. The shift from degenerate text to hallucinations is particularly concerning in the context of *scalable oversight* (Bowman et al., 2022; Kenton et al., 2024) making model uncertainty harder to detect.

To summarize, our main contributions are: (a) A controlled analysis of LLM behaviors under uncertainty, showing that strengthening models increases hallucinations over degenerate text and repetitions, (b) demonstrating that during generation, models undergo a phase shift from correct answers to hallucinations and then repetitions, (c) evidence that methods to reduce text degeneration, like sampling, increase errors and hallucinations. Our code and data is available at https://github.com/anonymous-submission933/fallbacks.

The rest of the paper is organized as follows: Section 2 discusses the different fallback behaviors in LLMs and defines naming conventions, while Section 3 outlines the experimental settings. The empirical results are divided into three parts: Section 4 presents results on the effect of modeling choices, such as parameter count and training procedures, on shifts in fallback behaviors; Section 5 shifts focus to the effect of inference hyperparameters, such as prompting templates and decoding methods, on the emergence of fallbacks; and lastly, Section 6 examines the order of appearance of different fallbacks within a single generation, with a particular focus on text degeneration.

## 2 FALLBACK BEHAVIORS

Since LLMs emerged as powerful generative tools (Radford et al., 2019; Brown et al., 2020), numerous reports have documented their unwanted behaviors. Holtzman et al. (2020) showed that greedy decoding methods can cause degenerate generations, such as incoherent text or repetitions. They propose the use of nucleus sampling to mitigate this problem. In another study, Nasr et al. (2023b) demonstrate that when given out-of-distribution inputs, models might output their training data verbatim. More recently, growing evidence (Zhu et al., 2024; Zhang et al., 2023; Aichberger et al., 2024) suggests that even the best models often generate convincing yet factually incorrect text, typically referred to as "hallucinations" or "confabulations" (Ji et al., 2022; Bang et al., 2023). Xiao & Wang (2021b) investigate model uncertainty, defining it as the probabilistic uncertainty in predicting the next token which is influenced by the model's training and knowledge, and identify it as a cause of hallucinations.[3]

In this paper, we propose to view these seemingly independent phenomena as fallback behaviors that models exhibit under uncertainty and investigate the relationships between them. Specifically, we focus on *epistemic uncertainty*, which refers to uncertainty due to a lack of knowledge (Hou et al., 2023). We hypothesize that the strength of an LLM influences its fallback behaviors, and aim to understand what factors determine how a model would behave in cases of uncertainty.

Our study considers the following phenomena: **1. Sequence repetition** When models face an inability to produce plausible continuations, they tend to repeat previously generated sequences, which are known to be plausible within the current context. **2. Degenerate text** As shown by Holtzman et al. (2020), models can generate degenerate text that is not strictly repetitive but follows a consistent pattern, such as enumerating or repeating sentences with variations in subject entities or attributes. **3. Hallucinations** We follow prior work in considering hallucinations as untruthful facts relative to the knowledge the model was exposed to during pretraining and the given context (Liu et al., 2022; Huang et al., 2023; Adlakha et al., 2023; Min et al., 2023a; Zhang et al., 2023). In cases that require the recollection of facts which the model cannot produce, it may fabricate coherent and seemingly plausible yet factually incorrect content.

We additionally looked for cases of verbatim recollection of training samples (Nasr et al., 2023a), but found no evidence of such fallback behavior in our setting (see further discussion in Section 6.1).

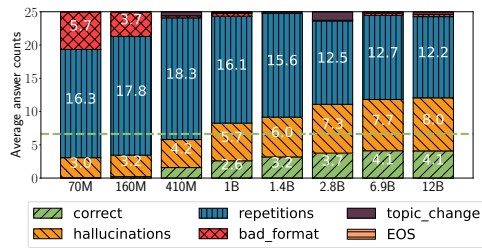

Figure 2: **Larger models resort to more complex fallback behaviors.** `Pythia` Models with larger parameter counts produce more correct facts (green) and hallucinations (orange) while less repeating facts (blue). The green line indicates the number of ground truth answers in TRIV.

## 3 EXPERIMENTAL SETTING

To directly investigate the relation between fallback behaviors and their contributing factors, our setup exposes models to naturally-occurring controlled cases of epistemic uncertainty. Specifically, we consider tasks that require recalling multiple facts about common topics, without engaging in complex reasoning. The tasks cover both structured outputs in the form of lists and open-ended generation, with queries spanning various difficulty levels, as explained next.

**Datasets** We use the following datasets:

1. **TRIVIAFACTS** (TRIV): We manually curate 95 high-quality domain-diverse questions with a list of answers (list questions) that (a) contains multiple elements, but no more than 15, (b) requires only knowledge that appears very frequently in the web, and (c) includes short elements without multiple correct phrasings. Due to its high-quality and highly controllable setup, we base the bulk of our experiments on this dataset, with minor modifications as relevant to each experiment.

2. **BIOGENERATION** (BIO): To investigate fallback behaviors in unstructured natural text, we follow Min et al. (2023a) and prompt the model to create biographies of entities from five popularity levels, analyzing the resulting facts. We sample 25 entities from each popularity level.

3. **QAMPARI** (QAMP): We sample 100 questions from the dataset introduced by Amouyal et al. (2023), which contains naturally occurring list questions with answers from Wikipedia.[4]

4. **FAKEQAMPARI** (FQAMP): We replace the subjects from QAMPARI with made-up names, verifying they do not exist. Introducing fictitious subjects reflects real-world queries with false premises (Brahman et al., 2024; Yuan et al., 2024), a common occurrence that forces models into epistemic uncertainty.

---

[3]While the model may experience internal uncertainty when predicting the next token, this uncertainty might not be reflected in the logits. Once a fallback behavior is selected, it may elevate certain tokens, leading to high logit values for these tokens in the final layers.

[4]Though similar to our TRIVIAFACTS dataset, this dataset has non-exhaustive answer sets and varying phrasings for the same answers, which can lead to correct answers being flagged as hallucinations.

Table 1 provides example questions, and additional details on the datasets are in Appendix B. Notably, unlike prior works that study model uncertainty by quantifying it (Xiao & Wang, 2021a; Lin et al., 2023; Zhang et al., 2024), here we introduce model uncertainty by construction. Namely, the above tasks require extensive recall of factual knowledge that is unambiguous and easy to evaluate, such that incorrect answers from the model must be attributed to this uncertainty.

**Generating predictions**    To remove behaviors caused by different decoding schemes, we perform our analyses with greedy decoding unless stated otherwise (Section 5.1 investigates the effects of temperature sampling). When requiring models to provide answers to the TRIV data, we prompt the model to produce a list of up to 25 answers (see Table 1 for example), thus forcefully pushing the models to recall facts, even when there are none. We ablate this behavior by removing the pre-defined length of the answer list and including specific demonstrations to complete the list while abstaining instead of generating incorrect facts (Section 5.2).

Notably, in some cases, we prompt the model to generate more facts than actually exist to observe its behavior. While seemingly synthetic, this setup mirrors common scenarios in natural LLM usage, where users prompt models to recall multiple facts, often on topics where the model may have knowledge gaps. These gaps arise from factors like knowledge cutoffs (Kadavath et al., 2022; Hou et al., 2023), false premises in prompts (Brahman et al., 2024), or queries about data not present in the pretraining set, such as proprietary or copyrighted material. We design our testbed to be diverse and representative, while remaining highly controllable, ensuring that epistemic uncertainty is the main cause of any incorrect answers produced by the model. This setup allows us to accurately classify each recalled fact into one of the possible fallback categories.

**Evaluation metrics**    Given a generated output, we parse it into a set of facts and evaluate each one as correct, repeated, or hallucinated. To avoid overestimating the model's factuality or hallucination rate, we classify only the first appearance of each fact as correct/hallucination, with all subsequent appearances treated as repetitions. For the list questions datasets (TRIV, QAMP and FQAMP), this parsing is mostly straightforward as the generations are structured as lists and the ground truth is given as a set of answers. For open-ended generation, we use FactScore Min et al. (2023a) to extract atomic facts and verify them against the entities' Wikipedia entries. As models frequently continue generating tokens after the completion of the instruction prompt, we further detect what the model did at that point. Namely, we consider the following options: 1) generating `EOS` token, 2) changing the topic, for example, by creating a new list/biography which we refer to as `topic change` or 3) continuing to predict tokens indefinitely (until the token budget is exhausted) within the same sentence/paragraph of the answer which we note by `bad format`. For additional information on the parsing process and example generations and endings, see Appendix C.

**Models**    We perform our experiments on a variety of model families, sizes, pretraining corpora sizes and finetuning stages. We evaluate both the base and chat-specific checkpoints of `Llama` 2[5] and `Llama` 3 which were finetuned on instruction and dialogue data  Touvron et al. (2023); Meta AI (2024). Secondly, we use `OLMo` Groeneveld et al. (2024), which comes in multiple model sizes, includes intermediate checkpoints throughout the pretraining phase and also offers instruction-tuned variants. Finally, we make use of the `Pythia` model family Biderman et al. (2023) which comes in 8 different scales, each with 154 intermediate pretraining checkpoints. We also include `Dolly` models which are instruction-tuned `Pythia` checkpoints (Conover et al., 2023). This suite of models allows us to control for different factors, such as pretraining data, model scale and training procedure, examining their effect on the emergence of fallback behaviors. We used FP16 checkpoints from HuggingFace for all models, except for `Llama` 2/3 70b which were loaded with FP8 precision.

## 4    FALLBACK BEHAVIORS OF DIFFERENT LLMS

> **Takeaway:** Increasing model strength through extra pretraining, more parameters, or instruction-tuning shifts fallback behaviors from simple to complex, i.e., from repetitions to hallucinations.

---

[5]We observed that `Llama` 2 13b, as released on HuggingFace, produces extremely poor results, leading us to suspect an incorrect upload of the weights. To avoid drawing incorrect conclusions based on a potentially defective model, we excluded it from our testbed. Further details and examples are provided in Appendix E.1.

Table 1: Example questions and answers for our main datasets.

* Taken from Wikipedia at `https://en.wikipedia.org/wiki/Harrison_Ford`.
♯ While not listed in Amouyal et al. (2023), Ryoichi Ikegami also wrote and drew *Spider-Man: The Manga*, according to `https://en.wikipedia.org/wiki/Ryoichi_Ikegami`.

| Dataset | Question | Example Answer |
|---------|----------|----------------|
| TRIV | The following 25 colors are in the Olympic rings\n 1. | Blue, Yellow, Black, Green, Red |
| BIO | The following is a bio of Harrison Ford:\n Harrison Ford | Harrison Ford (born July 13, 1942) is an American actor. He has been a leading man in … [*] |
| QAMP | The following 25 manga were drawn by Ryoichi Ikegami:\n 1. | Heat, Mai, the Psychic Girl, Wounded Man, Sanctuary, Crying Freeman, Strain[♯] |
| FQAMP | The following 25 manga were drawn by Haru Tanemura:\n 1. | *N/A* (Haru Tanemura is not a known manga creator) |

Scaling up models and training data improves performance and reduces undesired artifacts (Brown et al., 2020). Incorporating instruction-following phases aligns LLMs' outputs with human preferences (Ouyang et al., 2022). In this section, we investigate how these improvements influence model behavior under uncertainty. We first focus on the trade-off between sequence repetitions and hallucinations, measured as discrete shifts in fallback behaviors. Due to the various forms and broad definition of degenerate text, measuring its appearance is not straightforward. We revisit this in depth in Section 6, demonstrating that the fallback shift is a continuous rather than discrete process.

### 4.1 MODEL SCALE DICTATES THE FALLBACK BEHAVIOR

To minimize confounding factors and understand the direct effect of model scale on fallback behaviors, we generate predictions with greedy decoding over our TRIVIAFACTS data, and analyze the results by model family and scale. Since Pythia models were all trained on the same data in the same manner, they are the most comparable. Figure 2 shows that larger models recall more correct answers on average (green bar) and have lower failure rates in understanding task formats (red bar), as expected. However, a clear trend emerges: while smaller models struggle to recall many facts and resort to repeating the same ones (blue bar), as the number of parameters increases, repetitions are replaced with hallucinations (orange bar). This trend is consistent across the `OLMo` and `Llama` 2 model families (Figure 17). We also tested naturally occurring list questions from our QAMPARI subset, confirming the same trends (Figure 18).

Finally, we test what happens when we push uncertainty to the limit using our FAKEQAMPARI dataset, which has no correct answers. One might expect models to abstain, indicating they do not know the entity. However, as Figure 3 shows, not only do the models fail to abstain, but we also observe that larger and more advanced models are more likely to fabricate facts compared to their smaller counterparts. Specifically, the proportion of hallucinated answers more than doubles from approximately 15% in models with fewer than one billion parameters to over 30% in larger models, while the proportion of repetition decreases.

### 4.2 MODELS SHIFT FALLBACKS DURING PRETRAINING

Most LLMs are trained with autoregressive language modeling objective, maximizing the log probability of the next token given some context. Increasing the number of tokens seen during training lowers perplexity and improves language understanding (Kaplan et al., 2020). Using intermediate checkpoints from the `OLMo` and `Pythia` model families, we study fallback behavior during pretraining. Figure 4 depicts the fallbacks breakdown of `Pythia`-6.9B across training, showing that initially, after seeing only 2-4 billion tokens, it mainly repeats facts (blue). As training continues, it produces more correct answers (green) and more incorrect unique facts, i.e., hallucinations (orange), while repeating facts less. Similar trends were observed for `Pythia`-12B checkpoints, as well as for `OLMo` models (see Figures 19 and 20).

### 4.3 INSTRUCTION FINETUNING SHIFTS BEHAVIORS

Recently, instruction finetuning (Ouyang et al., 2022) has been adopted as a valuable method to improve model performance and align its generation with human preferences. Although one might assume such training reduces hallucinations, OpenAI et al. (2024) suggest it increases model miscalibration, resulting in hallucinations associated with high logit values.

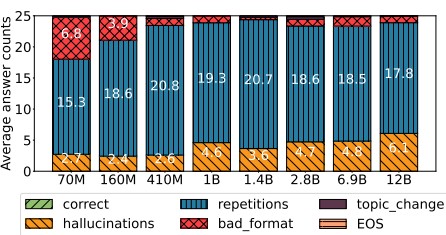 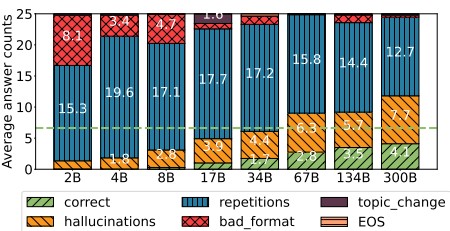

Figure 3: **Larger models hallucinate instead of abstaining.** When completing a list of facts about fictitious entities (FQAMP), larger Pythia models hallucinate more, while smaller models repeat facts. Models never abstain from giving incorrect facts.

Figure 4: **Models that train longer shift to complex fallbacks.** As Pythia-6.9B checkpoints see more training tokens (in billions), they produce more hallucinations and fewer repetitions. The green line shows correct answers upper bound in TRIV.

Repeating the experiments from Section 4.1 and comparing models to their instruction-tuned counterparts, we see a similar shift in fallback behavior; instruction-tuned models generate fewer repeated sequences and more hallucinations on average (Figures 21 and 22 in Appendix E depict the results of all instruction tuned models compared to their base versions on TRIVIAFACTS and QAMPARI, respectively). For the OLMo and Llama 2/3 family which had undergone more finetuning than Dolly checkpoints, the results are much more pronounced, with the hallucinations portion almost doubling between scales. One exception is Llama 3 70b which produced overall less facts, both correct , repetitions and hallucinations, opting to abstain through stopping the generation early or changing the topic. However, among the facts it it did provide, the hallucinations rate remained similar, with a slight increase from 48% for the base model to 52% in the chat version on TRIVIAFACTS and from 33% to 62% on QAMPARI. While pretrained LLMs are generally unable to generate sequences shorter than what they encountered during training, instruction-tuned models are more likely to produce an EOS token, preempting the generation early and resulting in fewer facts. However, we also note that while instruction finetuning can improve alignment with human preferences, it can sometimes cause finetuned models to diverge more frequently, resulting in some bad format generations.

## 4.4 SIMILAR TRENDS IN OPEN-ENDED GENERATION

To mimic real-world user requests, we use the BIOGENERATION dataset, sampling completions from a subset of the Pythia model scales with a temperature of $\tau = 0.5$. We use FactScore (Min et al., 2023b) to parse each generated biography into atomic facts and let ChatGPT 3.5 (Ouyang et al., 2022) verify them against Wikipedia entries.[6] Without the limitation on number of facts to produce, we observe that the larger models also generate more facts. Interestingly, when averaging over the *very-rare* entities (Figure 5) we see that even in this natural scenario, when models are required to elaborate on topics they know little about, they fall back to the same behaviors, with shifts occurring as predictably as in the controlled settings. Similar trends emerge over the rest of the popularity levels, though the more frequent an entity is, the more likely the models are to be able to recall facts for them and less uncertainty they face, thus making the results less useful for our analysis (Figures 34 to 38). This aligns only partially with Min et al. (2023b), who found stronger models generate more atomic facts and struggle more with rare entities. However, they observed that within a model family, larger models are generally more precise (i.e., hallucinate less). In contrast, Lin et al. (2022) found the largest models are the least truthful, which aligns with our findings.

## 5 FACTORS INFLUENCING THE FALLBACK BEHAVIORS OF AN LLM

> **Takeaway:** While LLMs have some internal capability to avoid hallucinations, this fallback behavior is inherent to their generation scheme and is likely unavoidable with current decoding methods. Mitigating degenerate text through random sampling often comes at the cost of more hallucinations.

---

[6]While FactScore, which relies on ChatGPT 3.5, can miss some atomic facts or incorrectly label them, we assume such errors occur at similar rates across generations, resulting in reliable trend analysis.

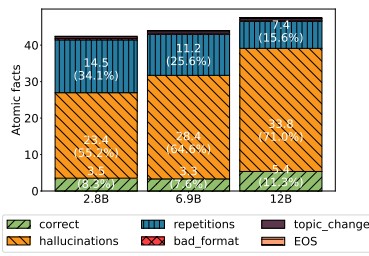

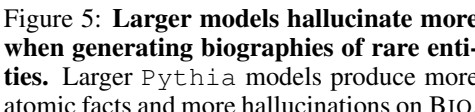

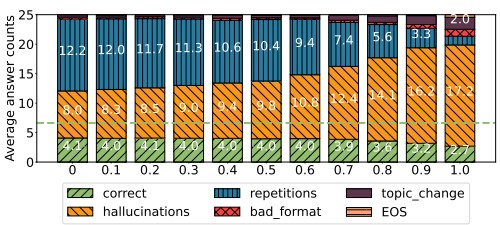

Figure 5: **Larger models hallucinate more when generating biographies of rare entities.** Larger `Pythia` models produce more atomic facts and more hallucinations on BIO.

Figure 6: **Increasing the sampling temperature shifts fallback behavior.** An average of 5 completions from `Pythia`-12B on TRIVI-AFACTS for varying temperatures.

In this section, we shift our focus from comparing different models to understanding the factors influencing the fallback behaviors of a frozen prertained model used for inference.

### 5.1 EFFECT OF SAMPLING METHODS

So far we mostly used greedy decoding to obtain responses from the models. However, in real-world applications, decoding often includes sampling from the model's output distribution and including repetition/frequency penalties (e.g., Holtzman et al., 2020; Keskar et al., 2019; Kumar et al., 2022). We analyze the effect of temperature sampling on fallback behaviors by repeating our experiment on TRIVIAFACTS, while generating five sequences per model for each input with an increasing temperature. Figure 6 shows the results for `Pythia`-12B, surprisingly revealing that while a higher temperature mitigates some repetitiveness, it causes a shift towards hallucinations. Moreover, introducing randomness reduces the number of correct facts,[7] which could be attributed to the random skipping of correct facts that have low confidence in the model. When repeating this experiment with additional models (Figures 29 to 33), as well as when using top-p ("nucleus") sampling (Holtzman et al., 2020), the results produce an identical trend. This further demonstrates the strong relationship between the fallback behaviors.

To assess whether our findings in Section 4.1 hold in realistic setups that involve random sampling, we repeat them with $\tau = 0.5$. This choice of temperature sets a good balance between the model's performance on the task and the amount of degenerate text it outputs (Figure 6). Figure 23 confirms that even with additional randomness, all fallback behaviors emerge, with shifts occurring as predictably as in the greedy-decoding case.

### 5.2 MODELS "VOLUNTEER" HALLUCINATIONS

In many previous experiments, we deliberately placed the model in high-uncertainty scenarios, prompting it to produce additional facts when none existed. To lift this restriction, we modify the base prompt in the TRIVIAFACTS data by removing the requirement for a specific number of answers (25). For example, the modification to the example in the first row of Table 1 would result in the prompt `''The following colors are in the Olympic rings:''`, allowing the model to stop generating additional answers once it exhausted its knowledge. The same trends persist: larger `Pythia` models reduce repetitions from 16.4 to 9.7 but increase hallucinations from 2.9 to 6.2 (Figure 24). This shows that even in natural scenarios without synthetic uncertainty, LLMs still rely on the same behaviors, failing to use better "exit strategies" like topic changes or abstention.

### 5.3 CAN FALLBACKS BE PREVENTED BY PROMPTING?

Kadavath et al. (2022) find encouraging evidence that LLMs may be calibrated (i.e. their confidence approximates the true probability of the output) and able to assess what they don't know, especially when they are larger. However, Kapoor et al. (2024a) recently showed that this observation does

---

[7]For example, if the first token for a correct fact has a probability of 0.51, it would be selected in greedy decoding but may be skipped with random sampling.

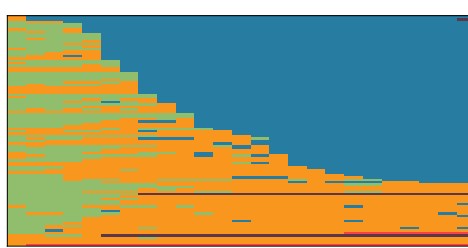

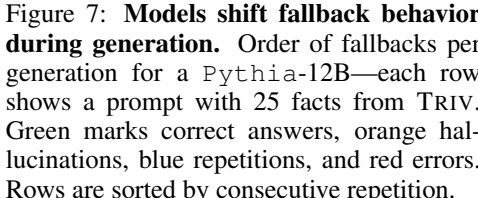

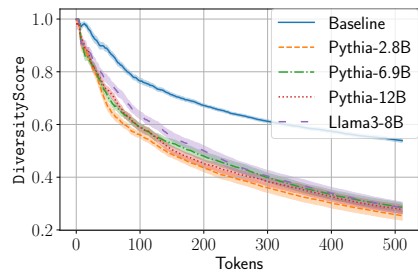

Figure 7: **Models shift fallback behavior during generation.** Order of fallbacks per generation for a `Pythia`-12B—each row shows a prompt with 25 facts from TRIV. Green marks correct answers, orange hallucinations, blue repetitions, and red errors. Rows are sorted by consecutive repetition.

Figure 8: **Model output gradually degenerates with increased generation length.** When generating biographies for rare entities, output degenerates as completion length increases compared to the human baseline (respective Wikipedia entries). Stronger models deteriorate more slowly.

not hold for some popular open-source models, and Yona et al. (2024) showed that LLMs struggle to express their uncertainty in words. We aim to understand whether LLMs are internally aware of their uncertainty, and if this awareness can reduce unwanted behaviors.

We modify the prompts as follows: For instruction-tuned models, we add the following prefix to each prompt `` `` `Complete the following list with facts you are sure of, and stop when you cannot recall additional facts` ''. For pretrained models, we add a prefix with three demonstrations for in-context learning, each consists of an easy list question and its corresponding answers followed by a topic-change to a new list (Figure 16 gives the full prefix).

The results for base models on TRIVIAFACTS and QAMPARI are presented in Figures 25 and 26, respectively. While for `Pythia` and `OLMo` we observe a minor increase in abstaining behavior at the expense of hallucinations, the overall trend remains the same, with hallucinated facts emerging abundantly (over 7 on average). In comparison, both `Llama` 2 and more pronouncedly `Llama` 3 manage to abstain by changing topics often (up to 20 facts on average), but the portion of hallucinated facts still increases with model size. Interestingly, when instructed to generate only facts that the model is certain about, nearly all instruction-tuned models generated even more hallucinations compared to their non-prompted counterparts, presumably indicating that they are "sure" of the facts they produce (Figures 27 and 28). Despite reacting differently to this modified instruction, all instruction-tuned models continued to exhibit similar fallback tendencies compared to when no explicit abstaining instruction was given. We conclude that fallback behaviors are inherent to current pretrained LLMs and emerge "unintentionally" and unavoidably under uncertainty.

## 6 FALLBACK BEHAVIORS IN ONE GENERATION

> **Takeaway:** As models generate longer texts, they shift in their fallback behavior, first generating hallucinations and eventually producing degenerate text.

While we established that both model strength and the decoding method impact the fallback behaviors of a model, these parameters are decided ahead of time. In this section, we focus on a single model at a time and investigate the effect of **generation length** on emergence of fallback behaviors.

### 6.1 EMERGENCE OF FALLBACKS DURING GENERATION

We view the facts generated by the model for a query as an ordered list of labels (*correct*, *hallucination*, *repetition*). For example, each row in Figure 7 shows the 25 labels of facts produced by `Pythia`-12B for each of the 95 samples in TRIVIAFACTS. Surprisingly, the model almost always first generates correct facts (green), then shifts to hallucinations (orange), and finally to repeating facts (blue). Notably, this fallback behavior isn't determined by the question and follows the same

order as when increasing model strength. This trend holds regardless of the model, the dataset, or the decoding method (see Figures 39 to 46 for further results).

To quantify this phenomenon, we denote a label of 1 for repetition, 2 for hallucination, and 3 for correct, and define

$$\texttt{ShiftScore}(f_1, \ldots, f_n) := \frac{1}{n-1} \sum_{i=1}^{n-1} \mathbb{1}_{f_{i+1} \geq f_i}$$

Where $f_1, \ldots, f_n$ is an ordered list of fact-labels ($f_i \in \{1, 2, 3\}$), and $\mathbb{1}$ is the indicator function. For each pair of dataset and model, we measure the $\texttt{ShiftScore}$ for each generation. We then run a Mann-Whitney U-test (McKnight & Najab, 2010) to compare the results to the expected $\texttt{ShiftScore}$ if the list of facts was produced in a random order (more details in Appendix E.2). We find that for all tested pairs, the p-value is $< 10^{-8}$ (see Table 2 for full analysis), and the U-statistic is always positive. We thus conclude that the ordering is not random, and follows the aforementioned hierarchy of fallback behaviors.

When investigating the fallback behaviors in a single generation, one may expect to observe occurrences of verbatim recollection of training samples, as suggested by Nasr et al. (2023a). To address this, we analyzed the generations from our experiments both qualitatively and quantitatively, and found no evidence of such behavior. We therefore hypothesize that verbatim generation of training samples stems from other factors than uncertainty regarding factual information, and is more prevalent in the presence of out-of-distribution inputs. For further details, see Appendix D.

### 6.2 FALLBACK SHIFTS IS A CONTINUOUS PROCESS

So far, we have analyzed model generations as discrete lists of facts, and correspondingly observed discrete shifts in fallback behaviors. In this section, we consider a softer measure of degenerate text generation. For example, producing a list of URLs such as *https://page.domain/**xyz***, *https://page.domain/**xyq***, *https://page.domain/**wyz*** is not strictly a repetition nor necessarily a hallucination, but it is clear that some form of conditional repetitiveness is occurring. Figure 12 shows another example as the model starts repeating previous facts (such as the shows Elsa Pataky appeared in) with a slightly different phrasing, before it begins repeating previous sequences verbatim. We consider *degenerate text* as sequences with repetitive textual patterns and/or rephrasing of previously generated text. Common patterns include enumerated lists of partially repeated items, repetitions of names and sentences with minor perturbations and permutation of previously generated statements. At it's peak, degenerate text is manifested by full verbatim repetition (Holtzman et al., 2020).

To measure such phenomena, we introduce a $\texttt{DiversityScore}$. Given a sequence of generated tokens $x = t_1, \ldots, t_n$ from the model's vocabulary $\mathcal{V}$, we define $\texttt{DiversityScore}(x) := \frac{1}{n} \sum_{v \in \mathcal{V}} \mathbb{1}_{v \in x}$, i.e. the number of unique tokens in the sequence divided by the sequence length. When $n \ll |\mathcal{V}|$ we expect $\texttt{DiversityScore} \approx 1$ while a repetitive sequence will have a score that diminishes rapidly towards 0.

Experimenting with human-written texts of $\leq 512$ tokens from Wikipedia, we observe they have a typical $\texttt{DiversityScore}$ of $0.7 - 0.8$ that almost never exceeds 0.6 (see Figure 8). Specifically, we use the subset of frequent entities in BIOGENERATION Min et al. (2023b), for which we expect models to be the least uncertain and generate full and diverse outputs, as discussed in Section 4.4. We compare the Wikipedia paragraphs for these entities with the model-generated biographies. For all models' outputs, the $\texttt{DiversityScore}$ diminishes much faster, with weaker models crossing the 0.5 threshold at around 150 tokens (i.e., after only 150 tokens, every other token appeared before). We observe a similar trend for the stronger model $\texttt{Llama}$ 3 8B (Meta AI, 2024).

We conclude that the shift in fallback behaviors towards the degenerate end does not occur in a discrete phase shift, but is a continuous process that exacerbates as the generation length grows.

## 7 RELATED WORK

**Repetitions and degenerate text** Holtzman et al. (2020) first attributed the tendency of LMs to generate highly repetitive and degenerate text to greedy sampling and suggested nucleus sampling as a possible mitigation. Follow-up work demonstrated the role of uncertainty in generating degenerate text and proposes various solutions using random, controlled, or constrained generation techniques

Keskar et al. (2019); Kumar et al. (2022); Zhang et al. (2022); Finlayson et al. (2023); Su et al. (2022); Li et al. (2023a). In a parallel approach, Olsson et al. (2022) focused on understanding the intrinsic mechanisms that cause models to copy previous inputs.

**Hallucinations**   As random sampling techniques became ubiquitous and LLMs grew in size and capability, the main focus shifted toward understanding the causes and proposing solutions to the generation of non-factual text, commonly referred to as hallucinations Ji et al. (2022); Huang et al. (2023). Recent work has focused on understanding how and why hallucinations emerge during generation Rashkin et al. (2023); Zhang et al. (2023); Adlakha et al. (2023); Kim et al. (2024), reducing hallucinations (e.g., by retrieval-augmentation or prompting) Roller et al. (2021); Shuster et al. (2021); Dhuliawala et al. (2023), and detecting hallucinations Zhou et al. (2021); Liu et al. (2022); Honovich et al. (2022); Min et al. (2023b); Mishra et al. (2024); Gottesman & Geva (2024); Hron et al. (2024). Recently, Denison et al. (2024) demonstrated how models generalize during training and learn to shift from simple dishonest strategies such as sycophancy to generating false facts with premeditation. Similarly, Band et al. (2024) suggest that hallucinating rather than abstaining is an issue with the calibration of the model, as was also hypothesized by OpenAI et al. (2024).

**Uncertainty in language modeling**   The phenomenon where LMs abstain or hallucinate when uncertain is well-known Xiao & Wang (2021b); Lin et al. (2022); Snyder et al. (2023); Baan et al. (2023); Kang et al. (2024). Recent studies have examined if the generation probability of LLMs is calibrated Kadavath et al. (2022) and whether they can express their uncertainty in natural language Li et al. (2023b); Yona et al. (2024). A parallel line of work focuses on defining notions of uncertainty and designing method to quantify it in various setups (Xiao & Wang, 2021a; Hou et al., 2023; Lin et al., 2023; Liu et al., 2024b; Ling et al., 2024). Unlike previous research, we focus on understanding model behaviors under *epistemic* uncertainty, rather on quantifying such uncertainty.

Specifically, we focus on epistemic uncertainty that reflects a lack of knowledge, regardless of its root cause, such as training data limitations, model architecture, or optimization strategies. This approach aligns with the definitions provided by Hüllermeier & Waegeman (2019), where epistemic uncertainty is characterized as reducible through additional data or better modeling, and complements the broader uncertainty taxonomy discussed by Baan et al. (2023). Additionally, Abbasi-Yadkori et al. (2024) emphasize the importance of distinguishing epistemic uncertainty from aleatoric uncertainty, which arises from inherent randomness, particularly in complex outputs or out-of-distribution settings. While our study primarily investigates epistemic uncertainty in knowledge-intensive use-cases of LLMs, it would also be valuable for future work to explore aleatoric uncertainty and its implications for model behavior in cases of ambiguous or out-of-distribution inputs.

## 8   CONCLUSION AND DISCUSSION

This work links the notorious unwanted behaviors of LLMs, such as degenerate and repetitive text and hallucinations, showing that they are all fallback behaviors models exhibit under uncertainty. We provide abundant evidence that these behaviors emerge with a clear ordering between their appearances, when comparing similar models of different strength, different decoding strategies and even in a single generation. Our experiments suggest that these fallback behaviors are inherent to current LLMs and that existing methods to alleviate them may simply replace one fallback by another. Moreover, longer training and additional parameters enhance performance and shift model fallbacks towards more complex range. However, as generation length grows, even the strongest models will resort to hallucinations and may eventually produce degenerate text.

Our findings have practical implications for the deployment and scalable oversight of LLMs (Bowman et al., 2022). Specifically, understanding fallback behaviors under epistemic uncertainty can inspire practitioners with actionable insights into how models respond when faced with gaps in knowledge. This is particularly critical as model sizes increase, given the observed tendency of larger models to exhibit subtler fallback behaviors, such as hallucinations, which are more challenging to detect compared to simpler repetitions. Future research may explore mechanisms to steer models toward predictable fallback behaviors, leveraging the connections between behaviors we identify to develop practical interventions for reducing hallucinations in deployed systems (Mosbach et al., 2024).

## REPRODUCIBILITY

We have made every effort to ensure the reproducibility of our work. All code, datasets, hyperparameters, and recipes required to reproduce our results and plots are available in our `https://github.com/anonymous-submission933/fallbacks`. This repository also contains all model generations and the complete set of plots, some of which are presented in the paper. None of the models or data used are proprietary; all experiments rely on open-source frameworks, including HuggingFace and PyTorch, and publicly available datasets. Detailed descriptions of the experimental setup and data processing steps can be found in the supplementary materials. Specifically, Section 3 outlines the experimental setup and the process for generating answers from various models. Appendix B describes the collection and processing procedures for each dataset used in the study. Additionally, Appendix C provides implementation details on the parsing process, including fact extraction and classification for each type of generated output.

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

APPENDIX

# A    LIMITATIONS

In this work we study different fallback behaviors of language models when faced with uncertainty. While we conduct multiple experiments trying to mimic real-world usage of such models as much as possible, there are several confounders that may still differentiate our controlled experiments than behavior in the wild. First, many of the commodity products are wrapped in additional levels of verification layers to reduce the effect of such behaviors, and it is possible that when observed as a whole, the behaviors of these products could differ significantly than their underlying language models.

Second, while we study the effect of random sampling, it becomes more common to apply even stricter modifiers to the language model next-word prediction distribution by using methods such as nucleus sampling, top-k decoding, repetition penalty and more (Holtzman et al., 2020; Keskar et al., 2019; Finlayson et al., 2023). We leave to future work to examine such decoding strategies on fallback behaviors.

Third, we study the effect of instruction-following finetuning on fallback behaviors, but it remains possible that additional preference alignment may allow models to be instructed not to hallucinate. While recent work (Kapoor et al., 2024b; Yona et al., 2024) has showed these models are not well calibrated and cannot tell whether they hallucinate or not, it is an open research and it is possible that eventually it will be a viable solution.

Finally, this work focuses on producing factually correct facts in natural language, and it remains for future work to investigate similar phenomena in producing faithful text to in-context information (such as entities position and attributes with a story), as well as when the task involves synthetic language such as when producing code.

# B    DATASETS

As discussed in Section 3, we make use of multiple datasets for our experiments. In this section, we provide further details on the construction of each of them. We release all the datasets used in our experiments at `https://github.com/anonymous-submission933/fallbacks`.

## B.1    TRIVIAFACTS

When creating this dataset, we set the following desiderata:

1. **Exhaustiveness**: In order to label all correct answers as such, we verify that our ground-truth answer set is exhaustive.

2. **Non-ambiguity**: To avoid incorrectly labeling a correct answer as a hallucination, we avoid questions where the answers may be phrased in many ways and focus on short answers with a single common way to refer to them.

3. **Easiness**: To be sure models are able to recall correct facts, we choose only questions where the answer list contains at least some answers that any graduate student can produce, as a proxy to the knowledge contained in language models that are trained on web data and Wikipedia.

4. **Diversity**: To avoid biases in evaluations, we set out to create a diverse set of questions that relates to as many domains as possible, spanning science, sports, culture, politics, geography, and more.

5. **Uncertainty**: To ensure questions induce uncertainty, we restrict the size of the ground-truth answer set to 10. As we ask models to produce 25 answers, we thus ensure they will become uncertain even when recalling all the correct answers.

We collect a set of nearly 300 candidate questions through a mix of manual suggestions and interactions with ChatGPT. We then manually annotate each of the questions according to the desiderata

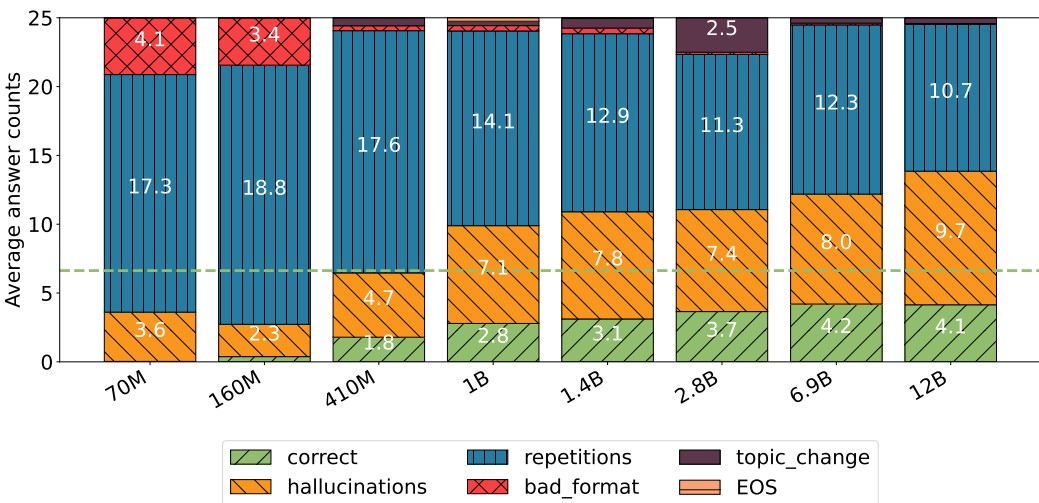

Figure 9: **Larger `Pythia` models' fallback behaviors on TRIVIAFACTS when adding a colon at the end of a questions.** Models with larger parameters count produce more correct facts (green) and hallucinations (orange) while less repeating facts (blue). The green line indicates the number of ground truth answers.

above, verifying the correctness and completeness of the answer set. After aggressive filtering, we are left with 95 high-quality questions that meet all of our requirements.

Table 1 shows the format of questions in TRIVIAFACTS. In contrast to the other list questions datasets (QAMPARI and FAKEQAMPARI), here the prompts do not end with a ":". To verify that this has no significant effect on the results, we append a colon to each question and repeat the experiments from Section 4.1. As depicted in Figure 9, the same trend repeat here, and the results are very similar to those in Figure 2.

### B.2 BIOGENERATION

Min et al. (2023b) introduce `FactScore`, which uses an LLM (e.g., ChatGPT) to parse an unstructured passage into atomic facts and verify them independently against a knowledge base. Min et al. (2023b) use topics from Wikipedia and divide them into five popularity levels based on their frequency in the knowledge base, from very rare to very frequent. They then require a model to generate an open-ended biography for each topic (entity) and use the respective Wikipedia page as the ground-truth knowledge source to verify facts. We randomly sample 25 topics from each of the five popularity levels and use them throughout our experiments.

### B.3 QAMPARI

We use the dataset as introduced by Amouyal et al. (2023). We filter the questions to those where the answer is a set of size $\geq 3$ and sample 100 random questions. Then, we rephrase each question to be in the format depicted in Table 1. Where appropriate, some of the questions follow a slightly different template. For example, one of the questions is *"Larry Cohen wrote and directed the following 25 works of art:"*.

### B.4 FAKEQAMPARI

We take the questions from QAMPARI (see Appendix B.3) and employ ChatGPT to replace the subject entity in each question with an alternative entity name which sounds plausible but does not exist (for example, choosing a Japanese name for an anime creator). We manually verified the generated questions, while checking that all new subjects feel plausible in the context and that they refer to entities that do not exist.

# C  PARSING ANSWERS

In this section, we provide further implementation details on the process used to parse the answers from each generation. We make our code and the model generations available in `https://gith ub.com/anonymous-submission933/fallbacks`.

## C.1  LIST QUESTIONS

**Extracting answers**  In Table 1, we show examples of inputs given to the model for completion. To steer models to complete the answers in a specified format, we append to each question the suffix ``\n 1.''. In almost all cases, the models indeed generate completions in the format of ``\n 1.  <answer 1>\n 2.  <answer 2> ...'', which allows us to easily extract the answers (Figure 10). In some cases, the generations include no \n or extra newline characters, but those cases are easily dealt with by a simple regex. The next most common case is a given answer as a comma-separated list. We identify those cases when none of the above options were detected and at least five commas appeared in the first sentence of the generation (no newlines or periods). Finally, if none of the above was detected but newline characters are identified (one or more) between short lines, we treat each new non-empty line as an answer. If none of the cases were identified, or a detected structured format is violated without new line characters, we collect the answers until that point and mark the difference from the expected 25 answers as missing due to `Bad Format`. All answers are normalized by removing extra white spaces to evaluate for repetitions.

**Identifying end of lists**  Since pretrained language models are often trained with sequences of a predetermined length, they do not produce end-of-sequence (`EOS`) tokens and continue to generate tokens until their budget is exhausted. As such, to avoid false-positive hallucination detection, we have to be able to identify when the model stopped producing answers for the given prompt and started generating completions for other topics. The easiest to identify and most common case is when the models generate the completion in the structured format as described above. In such cases, we can simply identify when the structure is violated (e.g., after the 10th answer the model stops enumerating answers and changes the topic), or take the first 25 answers given. Another common pattern used by many models is a `Topic change` by including the prefix "The following ..." (which we use as the prefix in our prompts), followed by a new list of items. Finally, some models (especially instruction-tuned ones) explicitly output an `EOS` token to mark the end of the generation, in which case we mark the missing answers to complete the expected 25 as missing due to `EOS`.

**Evaluating correctness**  To avoid labeling answers as hallucinations when they are correct but phrased differently, we perform a relaxed evaluation for correct answers. Namely, each given answer is normalized by removing redundant white spaces, articles, and punctuation, removing any parentheses, and making all characters lowercase. When multiple synonyms for the correct name exist (such as *Southern Ocean*, which is another name for *Antarctic Ocean*), we attempt to include both as possible correct answers. We then consider a produced answer correct if it has an $f_1$ score of at least $0.55$ with an answer in the ground-truth set.

We verify our methods manually by sampling generations from multiple models and experiments and find that the extraction, end-of-generation identification, and evaluation are correct in over $95\%$ of the cases.

## C.2  BIOGRAPHIES GENERATIONS

As we rely on FactScore (Min et al., 2023b) to extract atomic facts and evaluate them, our task here is mainly identifying topic changes to avoid false detection of hallucinations. To do so, we look for a prefix indicating a new biography is being generated, for special keywords such as *References*, *Discography*, etc., if multiple newline characters appear, or if a title-like line is encountered. We also consider any new line that starts with the subject name or a pronoun as a continuation of the biography.

After extracting only the part of the generation we consider as the biography and indicating the strategy to change the topic, we split the biography into sentences and delegate to FactScore (using ChatGPT 3.5 (Brown et al., 2020; Ouyang et al., 2022)) to extract atomic facts and validate them

```
The following 25 planets are in our solar system:
1.Mercury
2. Venus
3. Earth
4. Mars
5. Jupiter
6. Saturn
7. Uranus
8. Neptune
9. Pluto
10. Ceres
11. Pallas
12. Vesta
13. Juno
14. Vesta
15. Ceres
16. Pallas
17. Juno
18. Pallas
19. Vesta
20. Ceres
21. Pallas
22. Juno
23. Vesta
24. Ceres
25. Pallas

The following 25 planets are in our solar system
1. Mercury
2. Venus
3. Earth
4. Mars
5. Jupiter
6. Saturn
7. Uranus
8. Neptune
9. Pluto
10. Ceres
[... We omit the rest of the generation which continues to repeat the previous content indefinitely]
```

Figure 10: An example of a generation where the output followed the structured format precisely. For the extracted and labeled answer set, refer to Figure 11.

against the entity's Wikipedia page. An example of a full generation is given in Figure 12 and the list of extracted atomic facts and their labeling is given in Figure 13.

## D  HOW MODELS CHANGE TOPICS

Throughout our experiments, we analyze the classification of atomic facts in models' generations, up until a point where they change the topic. Nasr et al. (2023a); Geiping et al. (2024) demonstrate how extremely out-of-distribution inputs can cause models to fall back into recalling training samples verbatim, and Haviv et al. (2023); Stoehr et al. (2024) showed that generation of memorized sequences has a distinct internal "profile" from generation of non-memorized sequences. In contrast, Prashanth et al. (2024, inter alia) investigate memorization and find that it can be caused by different factors such as duplicated training data, repeating common and predictable patterns, and the recollection of sequences that are neither.

Inspired by this line of work, we analyzed how models change topics, and in particular, if memorized sequences emerge in these cases. We manually inspect such generations from multiple models, for both our TRIVIAFACTS and BIOGENERATION datasets, using both greedy and temperature sampling. Using Infini-gram (Liu et al., 2024a), we iterate over sliding windows of lengths 8 and 16 tokens to look for possible occurrences of verbatim recollection, which we manually inspect. We found no cases where a sequence longer than a few tokens repeats verbatim, and even then it comprises of common patterns, such as ``public static void main''.

Upon manual inspection, we have found that in almost all cases, the "topic change" included a repetition of the exact same content as before, additional data in some template format (e.g. listing references for BIOGENERATION samples), or another generation in the same format as the question (e.g. another list about another topic for TRIVIAFACTS and another biography for a different entity

```
Mars (C)
Uranus (C)
Venus (C)
Saturn (C)
Jupiter (C)
Mercury (C)
Neptune (C)
Earth (C)
Pluto (H)
Ceres (H)
Pallas (H)
Vesta (H)
Juno (H)
Vesta (R)
Ceres (R)
Pallas (R)
Juno (R)
Pallas (R)
Vesta (R)
Ceres (R)
Pallas (R)
Juno (R)
Vesta (R)
Ceres (R)
Pallas (R)"
```

Figure 11: Figure 10 shows an example of a generation where the output followed the structured format precisely. We display the extracted set of answers and their labeling according to our evaluation. An example breakdown of a generated biography into atomic facts and their labels is provided. Correct facts are noted by (C), hallucinated ones by (H), and repeated ones as (R).

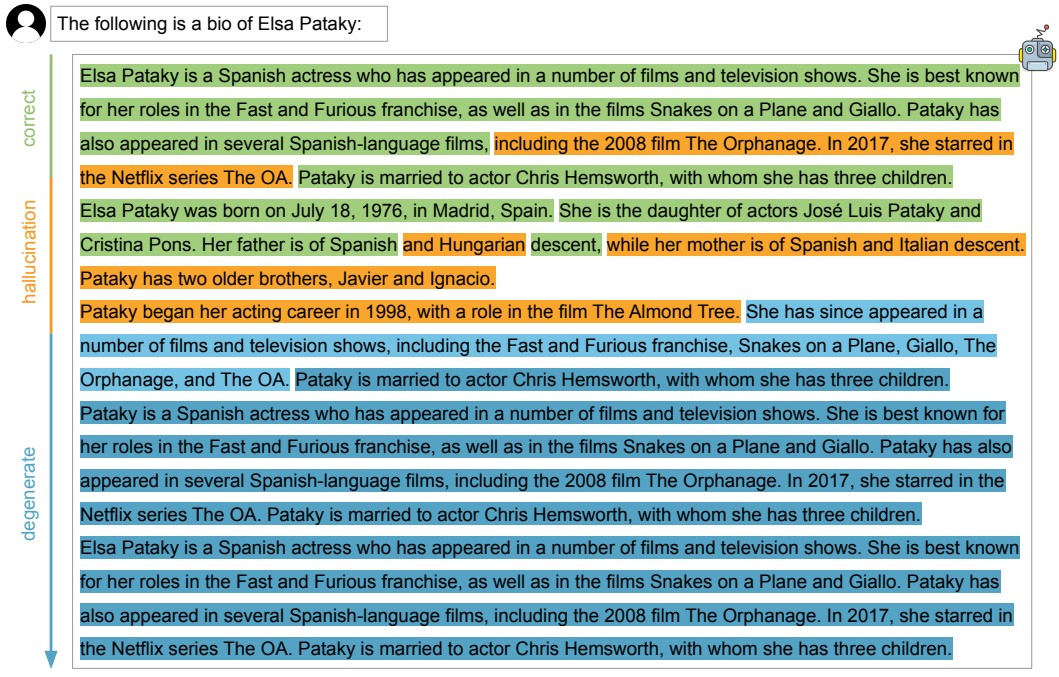

Figure 12: An example generated biography by `Llama` 3 8B, illustrating the different fallback behaviors of LLMs, gradually shifting from factually correct claims, through hallucinations and degenerate text, to sequence repetitions. The breakdown into atomic facts and their labels can be found in Figure 13

for BIOGENERATION). The only notable behavior observed was models attempting to provide some code snippets to support the answer-list generated, as seen in Figure 14.

```
Elsa Pataky is a Spanish actress. (C)
Elsa Pataky has appeared in a number of films. (C)
Elsa Pataky has appeared in a number of television shows. (C)
She is best known for her roles in the Fast and Furious franchise. (C)
She is best known for her roles in the films Snakes on a Plane. (C)
She is best known for her roles in the film Giallo. (C)
Pataky has appeared in several Spanish-language films. (C)
The Orphanage is a film. (C)
The Orphanage was released in 2008. (H)
In 2017, she starred in The OA. (H)
The OA is a Netflix series. (H)
Pataky is married to Chris Hemsworth. (C)
Pataky has three children. (C)
Pataky and Chris Hemsworth have three children together. (C)
Elsa Pataky was born on July 18, 1976. (C)
Elsa Pataky was born in Madrid. (C)
Elsa Pataky was born in Spain. (C)
She is the daughter of actors. (H)
Her father is José Luis Pataky. (H)
Her mother is Cristina Pons. (H)
Her father is of Spanish descent. (C)
Her father is of Hungarian descent. (H)
Her mother is of Spanish descent. (H)
Her mother is of Italian descent. (H)
Pataky has two older brothers. (H)
Pataky's older brothers are named Javier and Ignacio. (H)
Pataky began her acting career in 1998. (H)
Pataky's first role was in The Almond Tree. (H)
The Almond Tree is a film. (H)
She has appeared in a number of films. (C)
She has appeared in a number of television shows. (C)
The Fast and Furious franchise is a film. (C)
She has appeared in the Fast and Furious franchise. (C)
Snakes on a Plane is a film. (C)
She has appeared in Snakes on a Plane. (C)
Giallo is a film. (C)
She has appeared in Giallo. (C)
The Orphanage is a film. (H)
She has appeared in The Orphanage. (H)
The OA is a television show. (H)
She has appeared in The OA. (H)
Pataky is married to Chris Hemsworth. (R)
Pataky has three children. (R)
Pataky and Chris Hemsworth have three children together. (R)
Pataky is a Spanish actress. (C)
Pataky has appeared in films. (C)
Pataky has appeared in television shows. (C)
She is best known for her roles in the Fast and Furious franchise. (R)
She is best known for her roles in the films Snakes on a Plane. (R)
She is best known for her roles in the film Giallo. (R)
Pataky has appeared in several Spanish-language films. (R)
The Orphanage is a film. (R)
The Orphanage was released in 2008. (R)
In 2017, she starred in The OA. (R)
The OA is a Netflix series. (R)
Pataky is married to Chris Hemsworth. (R)
Pataky has three children. (R)
Pataky and Chris Hemsworth have three children together. (R)
Elsa Pataky is a Spanish actress. (R)
Elsa Pataky has appeared in a number of films. (R)
Elsa Pataky has appeared in a number of television shows. (R)
She is best known for her roles in the Fast and Furious franchise. (R)
She is best known for her roles in the films Snakes on a Plane. (R)
She is best known for her roles in the film Giallo. (R)
Pataky has appeared in several Spanish-language films. (R)
The Orphanage is a film. (R)
The Orphanage was released in 2008. (R)
In 2017, she starred in The OA. (R)
The OA is a Netflix series. (R)
Pataky is married to Chris Hemsworth. (R)
Pataky has three children. (R)
Pataky and Chris Hemsworth have three children together. (R)
(EOS)
```

Figure 13: An example breakdown of a generated biography into atomic facts and their labels. Correct facts are noted by (C), hallucinated ones by (H), and repeated ones as (R). (EOS) marks that the generation was preempted by the model, with no additional unrelated content generated. For the original generation as produced by the model, see Figure 12.

Overall, we did not observe any evidence of verbatim generation of training samples in the models' generations. We therefore hypothesize that this behavior stems from other factors than uncertainty regarding factual information, and is more prevalent in the presence of out-of-distribution inputs.

```
<OLMo 1B>
The following 25 are halogens in the periodic table\n1. Cesium\n2. Neon\n3. Fluorine\n4. Bromine\n5.
↪  Iodine\n6. Sulfur\n7. Phosphorus\n8. Chlorine\n9. Arsenic\n10. Mercury\n11. Cadmium\n12. Lead\n13.
↪  Tin\n14. Antimony\n15. Bismuth\n16. Tellurium\n17. Germanium\n18. Gallium\n19. Germanium\n20.
↪  Silicon\n21. Gallium\n22. Germanium\n23. Gallium\n24. Germanium\n25. Gallium\n\n## How to
↪  run\n\n```\n$ git clone https://github.com/julian-k/periodic-table.git\n$ cd periodic-table\n$
↪  python3 main.py\n```\n\n## License\n\nThis project is licensed under the MIT License - see the
↪  [LICENSE.md](LICENSE.md) file for details\n\n## Contributing\n\nPlease see
↪  [CONTRIBUTING.md](CONTRIBUTING.md)\n\n## Acknowledgments\n\nThis project is based on the
↪  [periodic-table](https://github.com/julian-k/periodic-table) project.

<Pythia 1B>
The following 25 are the cardinal directions on the compass rose\n1. North\n2. East\n3. South\n4.
↪  West\n5. Up\n6. Down\n7. Left\n8. Right\n9. Up\n10. Down\n11. Left\n12. Right\n13. Up\n14. Down\n15.
↪  Left\n16. Right\n17. Up\n18. Down\n19. Left\n20. Right\n21. Up\n22. Down\n23. Left\n24. Right\n25.
↪  Up\n\nA:\n\nI think you can use the following formula:\n\nNorth = (East + South) / 2\nEast = (West +
↪  South) / 2\nSouth = (North + West) / 2\n\nThe formula is:\n\nNorth = (East + South) / 2\nEast = (West
↪  + South) / 2\nSouth = (North + West) / 2\n\nThe formula is:\n\nNorth = (East + South) / 2...

<Pythia 6.9B>
The following 25 are phases of human sleep\n1. REM sleep\n2. NREM sleep\n3. Wakefulness\n4. Light
↪  sleep\n5. Deep sleep\n6. REM sleep\n7. NREM sleep\n8. Wakefulness\n9. Light sleep\n10. Deep
↪  sleep\n11. REM sleep\n12. NREM sleep\n13. Wakefulness\n14. Light sleep\n15. Deep sleep\n16. REM
↪  sleep\n17. NREM sleep\n18. Wakefulness\n19. Light sleep\n20. Deep sleep\n21. REM sleep\n22. NREM
↪  sleep\n23. Wakefulness\n24. Light sleep\n25. Deep sleep\n\nA:\n\nI think you are looking for
↪  this:\nimport java.util.Arrays;\nimport java.util.List;\n\npublic class Main {\n    public static
↪  void main(String[] args) {\n        List<String> list = Arrays.asList(\"1\", \"2\", \"3\", \"4\",
↪  \"5\", \"6\", \"7\", \"8\", \"9\", \"10\", \"11\", \"12\", \"13\", \"14\", \"15\", \"16\", \"17\",
↪  \"18\", \"19\", \"20\", \"21\", \"22\", \"23\", \"24\", \"25\");\n        System.out.println(list);\n
↪  }\n}\n\nOutput:\n[1, 2, 3, 4, 5, 6, 7, 8, 9, 10, 11, 12, 13, 14, 15, 16, 17, 18, 19, 20, 21, 22, 23,
↪  24, 25]\n\nA:\n\nYou can use a List<String> and use the index to get the value.\nList<String> list =
↪  new ArrayList<String>();\nlist.add(\"1\");\nlist.add(\"2\");
↪  \nlist.add(\"3\");\nlist.add(\"4\");...
}
```

Figure 14: A few examples of generation with greedy decoding. $< \cdot >$ indicate the generating model, and \n indicate line breaks in the generation. We use ``...'' when the sequence becomes repetitive.

# E  ADDITIONAL RESULTS

In this section, we provide additional results to support the results discussed in Sections 4 to 6. As discussed in Section 3, we evaluate three model families over multiple datasets and in various settings. For the sake of brevity, we include here a representative set of results and share the code and results to produce all plots for the rest of the experiments.

Figures 17 and 18 depict the scaling behavior of larger models on the TRIVIAFACTS and QAMPARI datasets respectively. Figures 19 and 20 shows fallback trends during pretraining for Pythia-6.9B and 12B models in the former and OLMo-1B and 7B models in the latter. Figures 21 and 22 gives similar results when adding instruction finetuning to models on the TRIVIAFACTS and QAMPARI datasets. In Figures 27 and 28 we ablate the prompt for instruction finetuned models to nudge them into abstaining rather than producing hallucinations on TRIV and QAMP respectively, and Figures 25 and 26 uses in-context examples to do the same for base models. Section 5.1 shows how additional randomness in sampling reduces repetitions, and trades it with hallucinations (Figure 6. Figures 29 to 33 provide additional evidence for these phenomena, repeating the experiments both with additional models, as well as using top-p sampling. Finally, Figures 34 to 38 bring a breakdown of the results on the BIOGENERATION datasets with different levels of popularity.

## E.1  EXCLUSION OF LLAMA 2 13B CHECKPOINT

We find that Llama 2 13b as released in meta-llama/Llama-2-13b-hf almost always produces extremely poor results. Upon manual inspection of its outputs, we conclude that there is likely a bug in the uploaded weights of this specific model, and thus we exclude it. We provide a few examples in Figure 15. We note that for the other checkpoints, as well as the chat variants of all three sizes, the generation are considerably of higher quality, further supporting this decision.

```
The following 25 fingers are part of the human hand:
    1. 1. 1. 1. 1. 1. 1. 1. 1. 1. 1. 1. 1. <this continues indefinitely>

The following 25 are signs of the Zodiac:
1. *The* *Sword* *of* *the* *Lictor*
2.  *The* *Sword* *of* *the* *Lictor*
3.  *The* *Sword* *of* *the* *Lictor* <this list continues indefinitely>

The following 25 are athletic events in the Decathlon for men:
1.  *

     *

     *  <this empty itemized lise continues indefinitely>

The following 25 are seasons of the year:
1. *The History of the Decline and Fall of the Roman Empire*
   1. Volume 1
   2. Volume 2
   3. Volume 3 <this list continues indefinitely>
```

Figure 15: A few examples of generation with greedy decoding from the `meta-llama/Llama-2-13b-hf` checkpoint. **Bold text** indicates the prompt. **Boldface text in** <> indicates authors' notes. This model often produces incoherent text, leading us to conclude there is a bug in the uploaded weights.

```
The following 25 are known moons of Mars
1. Phobos
2. Deimos

The following 25 are the species of the main characters with a dialogue in the movie The Lion King
1. Lion
2. Warthog
3. Meerkat
4. Mandrill
5. Hyena
6. Hornbill

The following 25 are the vegetables common in traditional greek salads
1. Tomato
2. Cucumber
3. Onion
4. Pepper
5. Kalamata olive
```

Figure 16: Prefix given to pretrained language model before each sample of the TRIVIAFACTS dataset to encourage in-context learning and demonstrate how topic-change can be done, half-way through a list, when the full answer set is exhausted, for the experiments mentioned in Section 5.2

### E.2 ORDER OF FACTS IN A SINGLE GENERATION

Section 6.1 introduces the `ShiftScore`, which measures how predictable the order of facts is with respect to the hierarchy between fallback behaviors as introduced in this work. To perform the Mann-Whitney U-test, we consider only answer sets with at least five unique answers and model-dataset pairs with at least 30 such answer sets. For each such set, we compute the expected `ShiftScore` of a random ordering of the answers by taking 1000 random permutations of their order and averaging their `ShiftScore`.[8] We then perform the statistical test on the list of `ShiftScore` values from the original ordering of the model against the scores of the random orders. Table 2 shows the p-value of the two lists of scores coming from the same distribution. In all cases, the U-statistic was strictly positive, allowing us to conclude that the original ordering follows the expected order in a statistically significant way.

Additionally, in Section 6.1 we discuss the ordering of fact-labels within a single generation, as presented in Figure 7 for `Pythia`-12B model on TRIVIAFACTS. Figures 39 to 48 give similar plots for other models.

---

[8]For TRIVIAFACTS with temperature sampling, we use the ordering of the first of the five generations sampled for each question.

Table 2: The results (p-values) of running Mann-Whitney U-test on the ordering of facts with respect to the predicted hierarchy introduced in this paper, for each model-dataset pair. In all cases, the p-value is less than $10^{-8}$ while the U-statistic is strictly positive suggesting the order of facts is far from being random.

| Model | QAMPARI | TRIV | TRIV ($\tau = 0.5$) |
|---|---|---|---|
| Pythia-70M | $1.7 \times 10^{-12}$ | $2.2 \times 10^{-13}$ | $7.8 \times 10^{-11}$ |
| Pythia-160M | $1.4 \times 10^{-18}$ | $2.0 \times 10^{-12}$ | $2.3 \times 10^{-15}$ |
| Pythia-410M | $4.8 \times 10^{-22}$ | $3.2 \times 10^{-20}$ | $1.6 \times 10^{-12}$ |
| Pythia-1B | $1.3 \times 10^{-18}$ | $7.8 \times 10^{-23}$ | $1.5 \times 10^{-16}$ |
| Pythia-1.4B | $1.5 \times 10^{-17}$ | $1.0 \times 10^{-25}$ | $1.4 \times 10^{-17}$ |
| Pythia-2.8B | $8.4 \times 10^{-16}$ | $2.1 \times 10^{-22}$ | $1.7 \times 10^{-19}$ |
| Pythia-6.9B | $3.3 \times 10^{-14}$ | $3.8 \times 10^{-28}$ | $1.1 \times 10^{-17}$ |
| Pythia-12B | $8.0 \times 10^{-19}$ | $9.8 \times 10^{-28}$ | $1.1 \times 10^{-22}$ |
| OLMo-1B | $4.9 \times 10^{-15}$ | $8.3 \times 10^{-22}$ | $2.4 \times 10^{-21}$ |
| OLMo-7B | $7.4 \times 10^{-20}$ | $3.8 \times 10^{-21}$ | $9.2 \times 10^{-24}$ |
| Llama2-7B | $8.7 \times 10^{-11}$ | $9.2 \times 10^{-26}$ | $6.2 \times 10^{-18}$ |
| Llama2-70B | $5.1 \times 10^{-09}$ | $4.3 \times 10^{-18}$ | $1.4 \times 10^{-12}$ |
| Llama3-8B | $2.2 \times 10^{-13}$ | $2.1 \times 10^{-11}$ | $1.4 \times 10^{-12}$ |
| Llama3-70B | $8.2 \times 10^{-10}$ | $3.6 \times 10^{-11}$ | $7.8 \times 10^{-11}$ |

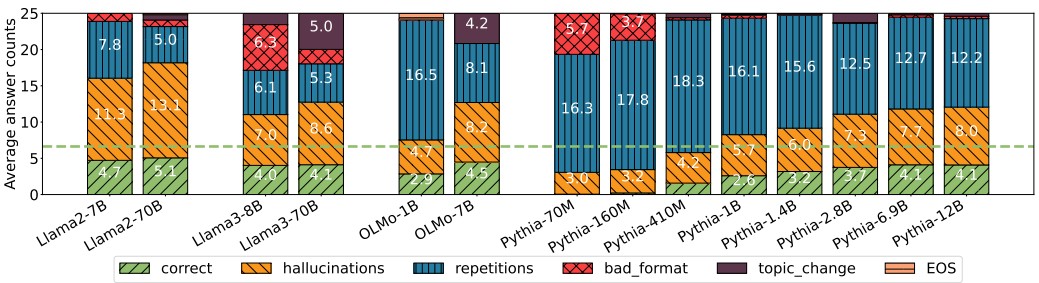

Figure 17: **Larger models resort to more sophisticated fallback behaviors on the TRIVIAFACTS dataset.** Here, models of increasing size produce more correct facts (green) and hallucinations (orange) while producing fewer repeated facts (blue). The horizontal green line indicates the maximum number of correct answers possible.

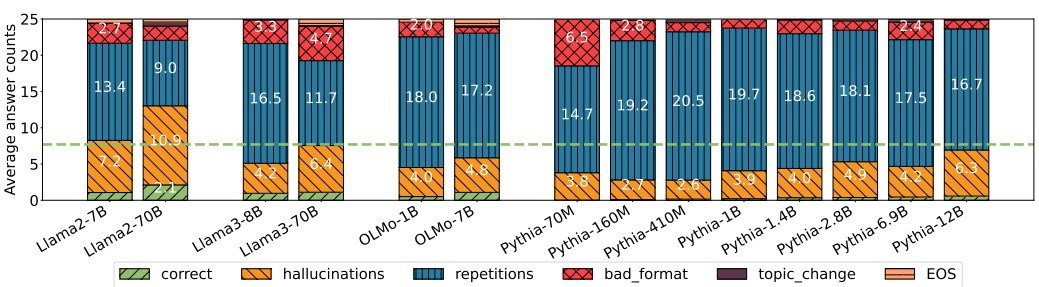

Figure 18: **Larger models resort to more sophisticated fallback behaviors on the QAMPARI dataset.** Here, models of increasing size produce more correct facts (green) and hallucinations (orange) while producing fewer repeated facts (blue). The horizontal green line indicates the maximum number of correct answers possible.

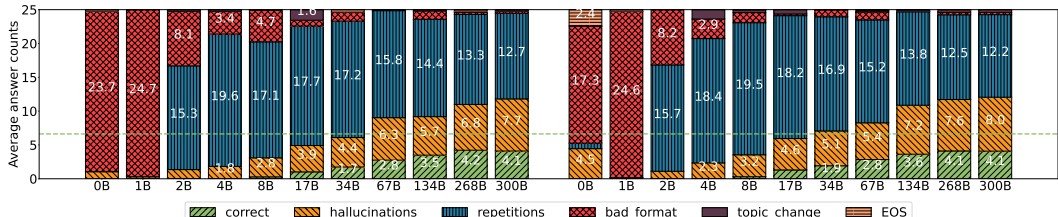

Figure 19: **Pythia models that train longer shift to complex fallbacks.** The more training tokens Pythia models see (in billions), the more hallucinations they produce and the fewer repetitions they generate (on TRIVIAFACTS). The left group depicts the trend for Pythia-6.9B pretraining while the right group is for Pythia-12B. The horizontal green line indicates the maximum number of correct answers possible.

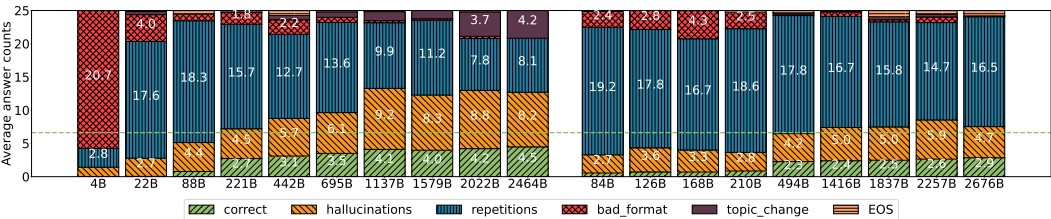

Figure 20: **OLMo models that train longer shift to complex fallbacks.** The more training tokens Pythia models see (in billions), the more hallucinations they produce and the fewer repetitions they generate (on TRIVIAFACTS). The left group depicts the trend for OLMo-1B pretraining while the right group is for OLMo-7B. The horizontal green line indicates the maximum number of correct answers possible.

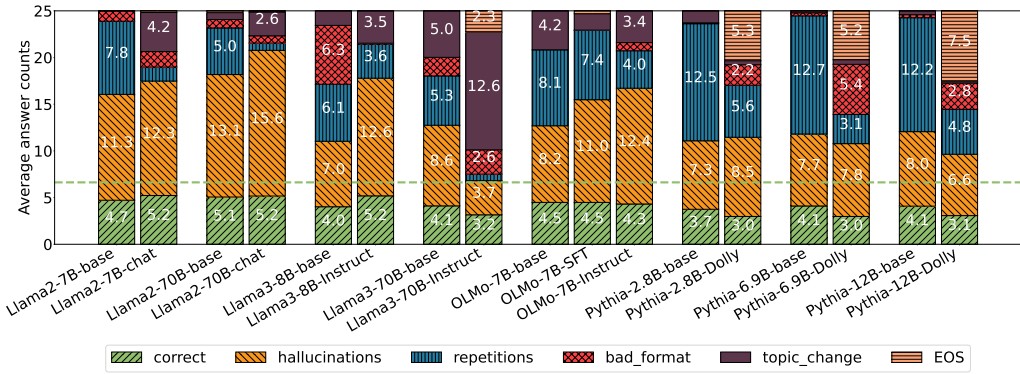

Figure 21: **Instruction-tuned models resort to more complex fallback behaviors, on the TRIV-IAFACTS dataset.** The Dolly models (instruction-tuned variants of `Pythia`) hallucinate more and repeat facts less, while also breaking out of loops more often and abstaining from producing additional facts (increase in red and pink bars). For the `OLMo` and `Llama 2` families, which had a more robust phase of instruction-following training, the results are much more pronounced. The horizontal green line indicates the maximum number of correct answers possible.

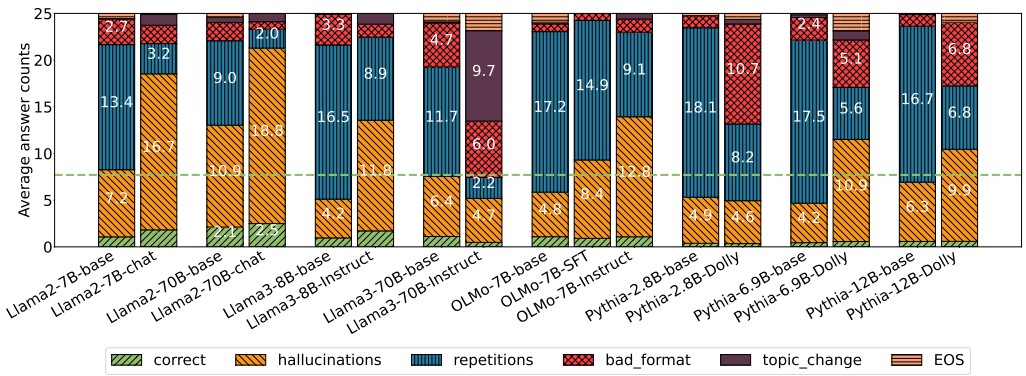

Figure 22: **Instruction-tuned models resort to more complex fallback behaviors, on the QAM-PARI dataset.** The Dolly models (instruction-tuned variants of `Pythia`) hallucinate more and repeat facts less, while also breaking out of loops more often and abstaining from producing additional facts (increase in red and pink bars). For the `OLMo` and `Llama 2` families, which had a more robust phase of instruction-following training, the results are much more pronounced.

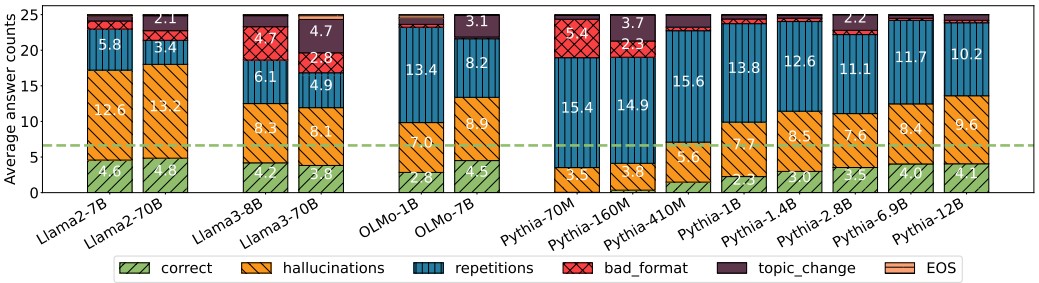

Figure 23: **Larger models use more sophisticated fallbacks even with random decoding.** Results of different models on the TRIVIAFACTS dataset with temperature sampling, setting the sampling temperature ($\tau$) to 0.5. Each completion was sampled five times. The horizontal green line indicates the maximum number of correct answers possible. The horizontal green line indicates the maximum number of correct answers possible.

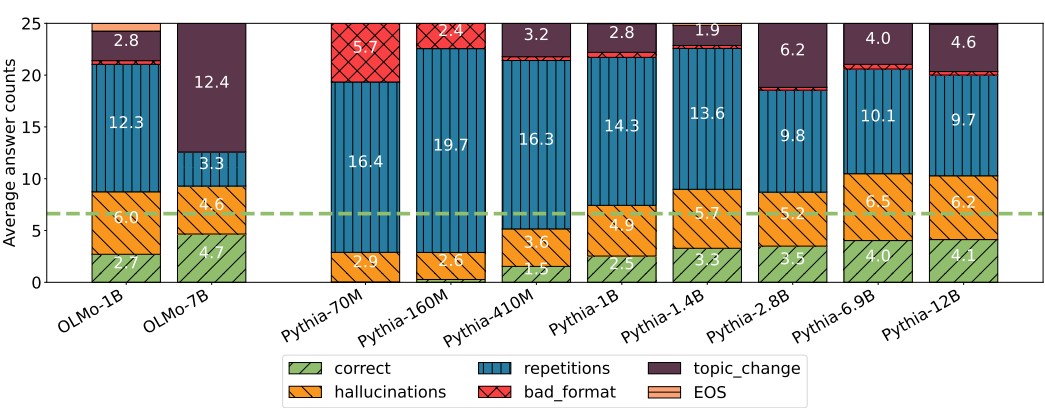

Figure 24: **Larger models volunteer hallucinations, even when not asked for additional facts.** When given prompts from TRIVIAFACTS without specifying the number of items in advance (see Section 5.2), larger models continue to produce more hallucinations than their smaller counterparts and barely exhibit abstaining strategies (e.g., changing the topic). The horizontal green line indicates the maximum number of correct answers possible.

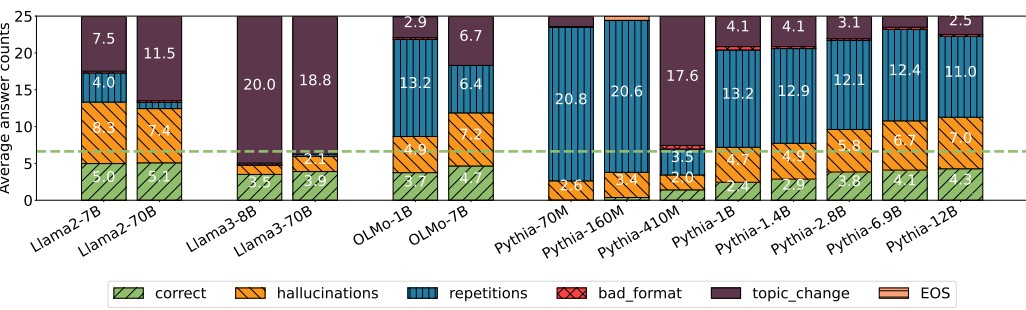

Figure 25: **Larger models hallucinate more on TRIVIAFACTS, even when shown how to abstain.** When given the prefix from Figure 16 to encourage abstaining instead of hallucinating, larger Pythia and OLMo models continue to exhibit the same fallback behavior trends, with repetitions decreasing and hallucinations increasing as model size grows. The models are only slightly more inclined to abstain (by changing topics). Notably, while Llama 2 and Llama 3 models exhibit the predicted trends in the proportion of hallucinations, they are considerably more capable of abstaining when given the opportunity. Interestingly, Pythia-410M is able to change topics remarkably well, though when manually inspecting its outputs, we find that in almost all cases it produces a list of five facts, often with repetitions, and then continues to repeat this list indefinitely. The results are given for the TRIVIAFACTS dataset, and the horizontal green line indicates the maximum number of correct answers possible.

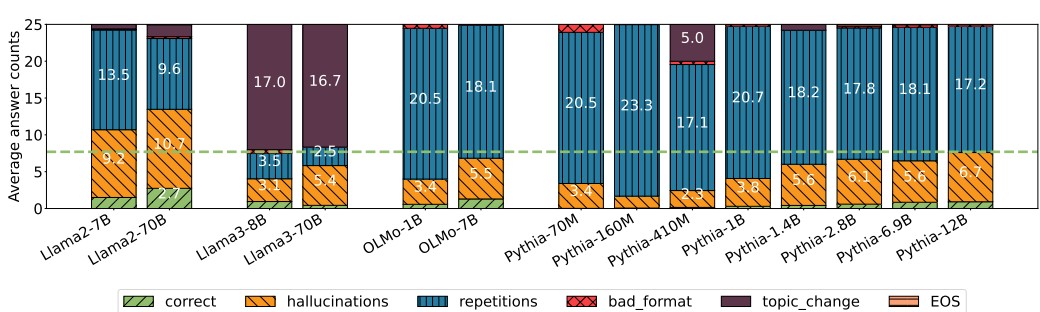

Figure 26: **Larger models hallucinate more on QAMPARI, even when shown how to abstain.** When given the prefix from Figure 16 to encourage abstaining instead of hallucinating, larger `Pythia` and `OLMo` models continue to exhibit the same fallback behavior trends, with repetitions decreasing and hallucinations increasing as model size grows. The models are only slightly more inclined to abstain (by changing topics). Notably, while `Llama` 2 and `Llama` 3 models exhibit the predicted trends in the proportion of hallucinations, they are considerably more capable of abstaining when given the opportunity. Similarly to the phenomana observed in Figure 16, `Pythia`-410M is able to change topics more than expected, though upon manually inspection of its outputs, we find that in almost all cases it produces a list of just a few facts, often with repetitions, and then continues to repeat this list including the prompt indefinitely. The results are given for the QAMPARI dataset, and the horizontal green line indicates the maximum number of correct answers possible.

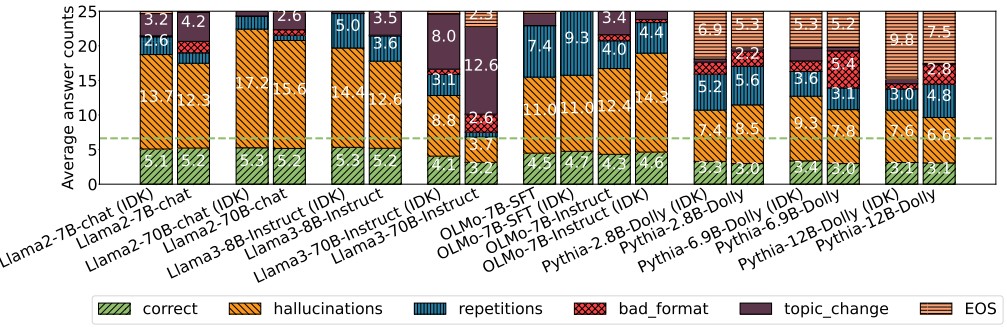

Figure 27: **Instruction tuned models continue to hallucinate on TRIVIAFACTS, even when instructed to generate only facts that the model is certain about.** When given the prefix `''Complete the following list with facts you are sure of, and stop when you cannot recall additional facts''` to encourage abstaining instead of hallucinating, all instruction tuned models ( "(IDK)" marking the model with the modified prompt) fail to utilize internal uncertainty estimations if exists and continue to hallucinate. Some models are slightly more inclined to abstain (by changing topics or producing `EOS` tokens), while others attempt to compete the list even more than before resulting in additional hallucinations. The horizontal green line indicates the maximum number of correct answers possible.

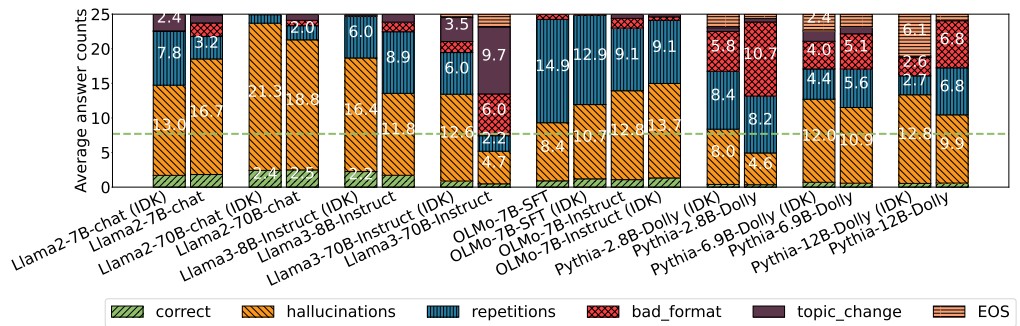

Figure 28: **Instruction tuned models continue to hallucinate on QAMPARI, even when instruction to generate only facts that the model is certain about.** When given the prefix ``Complete the following list with facts you are sure of, and stop when you cannot recall additional facts'' to encourage abstaining instead of hallucinating, all instruction tuned models ( "(IDK)" marking the model with the modified prompt) fail to utilize internal uncertainty estimations if exists and continue to hallucinate. Some models are slightly more inclined to abstain (by changing topics or producing EOS tokens), while others attempt to compete the list even more than before resulting in additional hallucinations. The horizontal green line indicates the maximum number of correct answers possible.

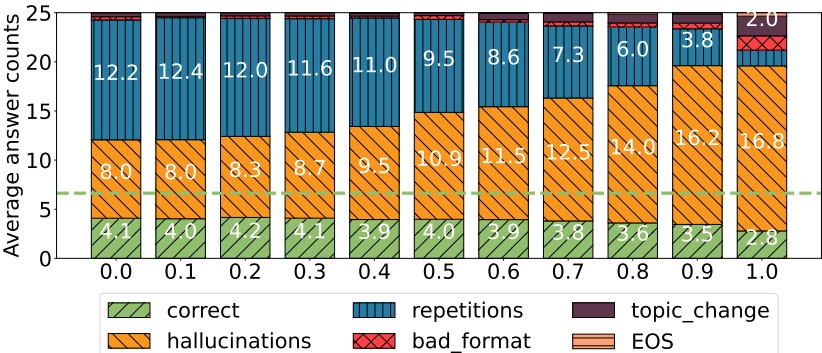

Figure 29: **Using larger $p$ with top-p sampling shifts fallback behavior from repetitions to hallucinations.** An average of 5 completions from Pythia-12B on TRIVIAFACTS for varying temperatures.

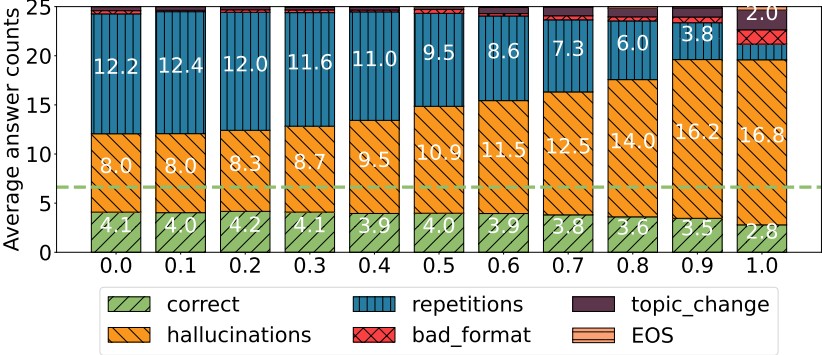

Figure 30: **Using larger $p$ with top-p sampling shifts fallback behavior from repetitions to hallucinations.** An average of 5 completions from Pythia-6.9B on TRIVIAFACTS for varying temperatures.

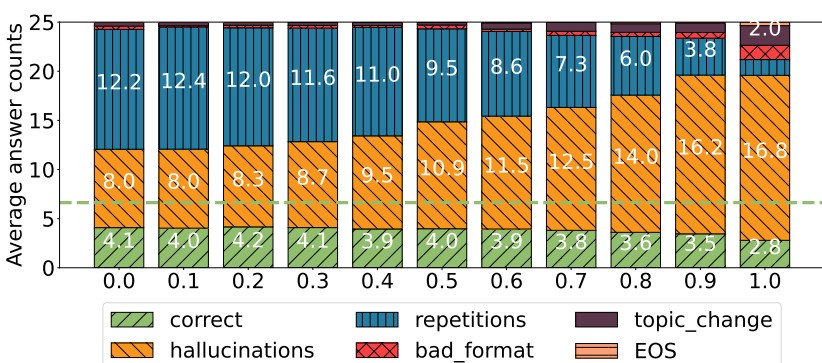

Figure 31: **Using larger $p$ with top-p sampling shifts fallback behavior from repetitions to hallucinations.** An average of 5 completions from `Pythia`-2.8B on TRIVIAFACTS for varying temperatures.

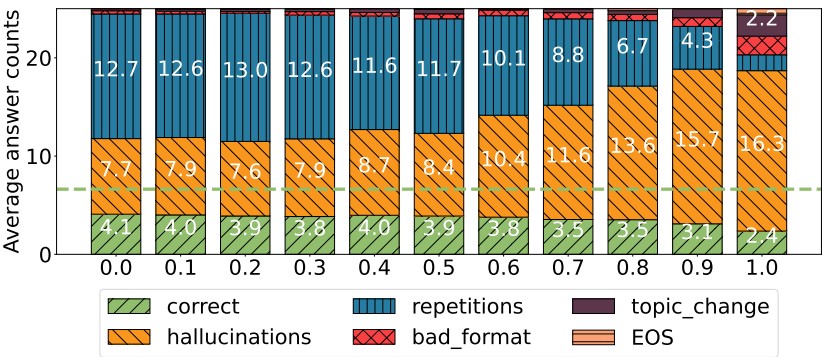

Figure 32: **Increasing the sampling temperature shifts fallback behavior.** An average of 5 completions from `Pythia`-6.9B on TRIVIAFACTS for varying temperatures.

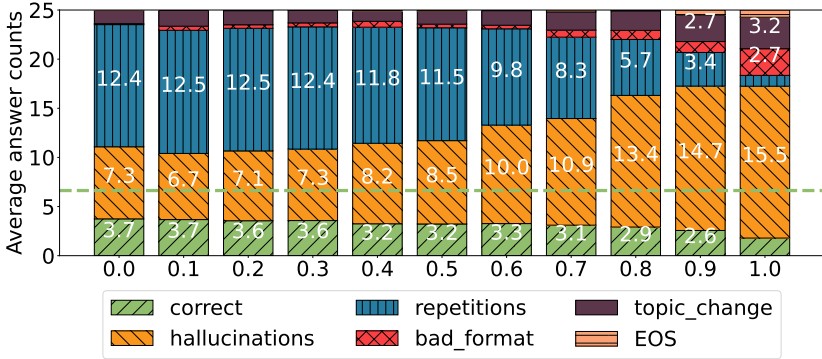

Figure 33: **Increasing the sampling temperature shifts fallback behavior.** An average of 5 completions from `Pythia`-2.8B on TRIVIAFACTS for varying temperatures.

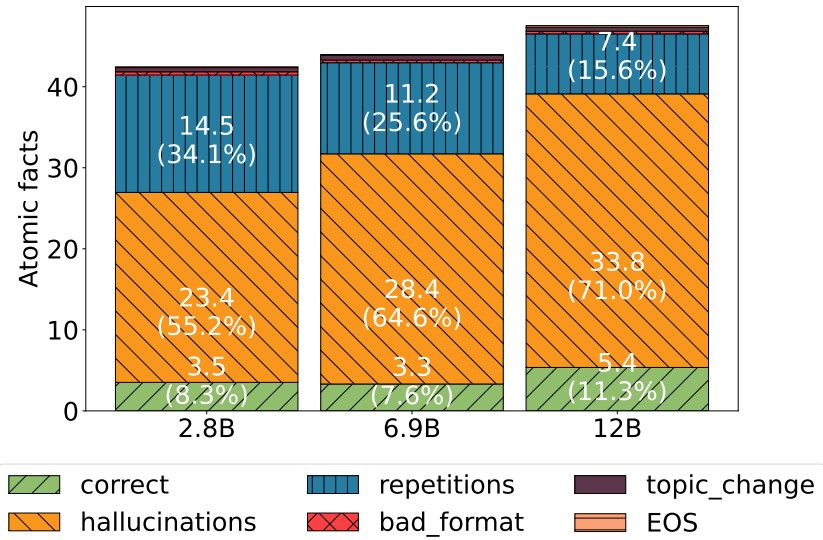

Figure 34: **Larger `Pythia` models hallucinate more when generating open-ended biographies on very rare entities.** When producing biographies of entities with very low frequency of appearance in Wikipedia, larger models generate more atomic facts, with an increasing rate of hallucinations.

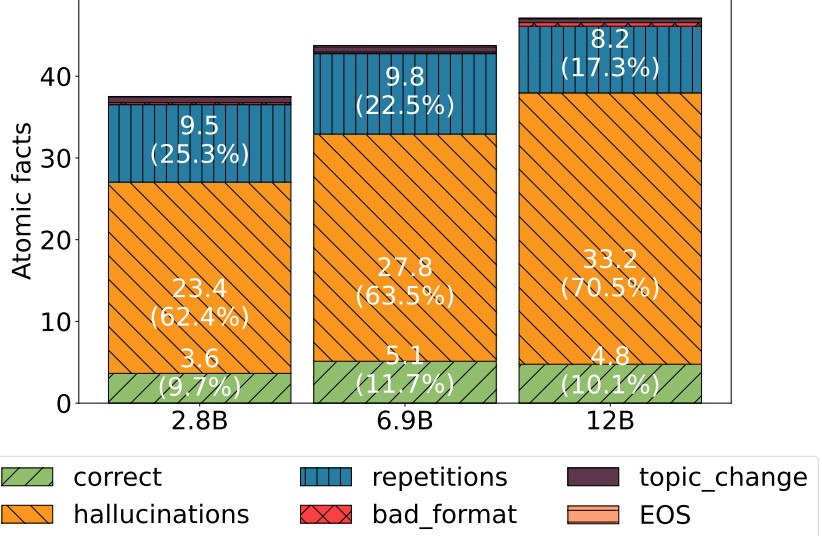

Figure 35: **Larger `Pythia` models hallucinate more when generating open-ended biographies on rare entities.** When producing biographies of entities with low frequency of appearance in Wikipedia, larger models generate more atomic facts, with an increasing rate of hallucinations.

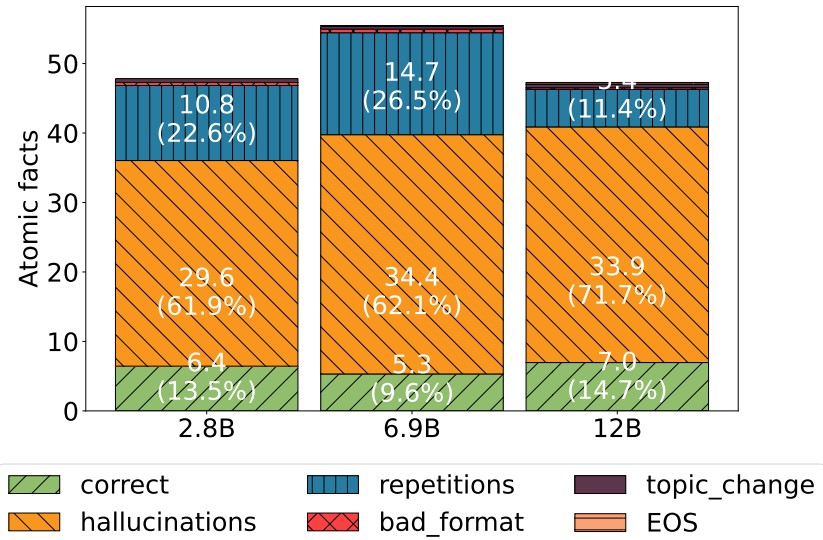

Figure 36: **Larger `Pythia` models hallucinate more when generating open-ended biographies on medium-popularity entities.** When producing biographies of entities with medium frequency of appearance in Wikipedia, larger models generate more atomic facts, with an increasing rate of hallucinations.

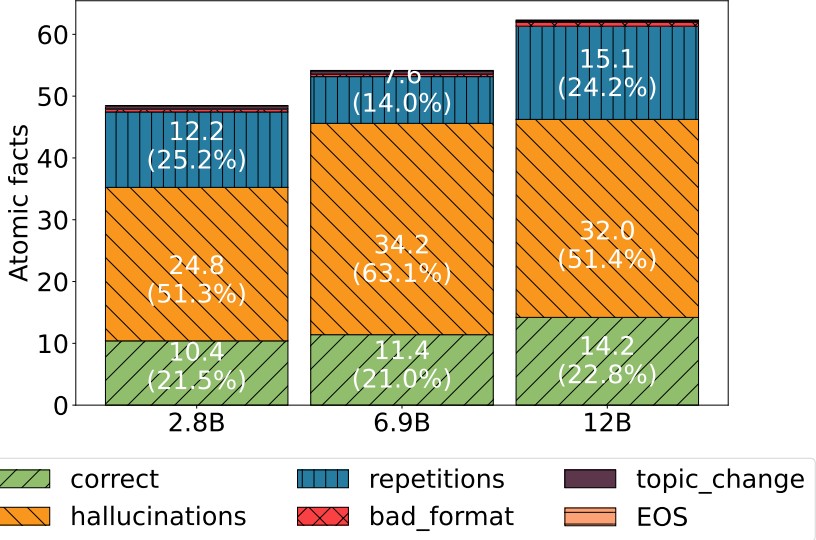

Figure 37: **Larger `Pythia` models hallucinate more when generating open-ended biographies on popular entities.** When producing biographies of entities with high frequency of appearance in Wikipedia, larger models generate more atomic facts, with an increasing rate of hallucinations.

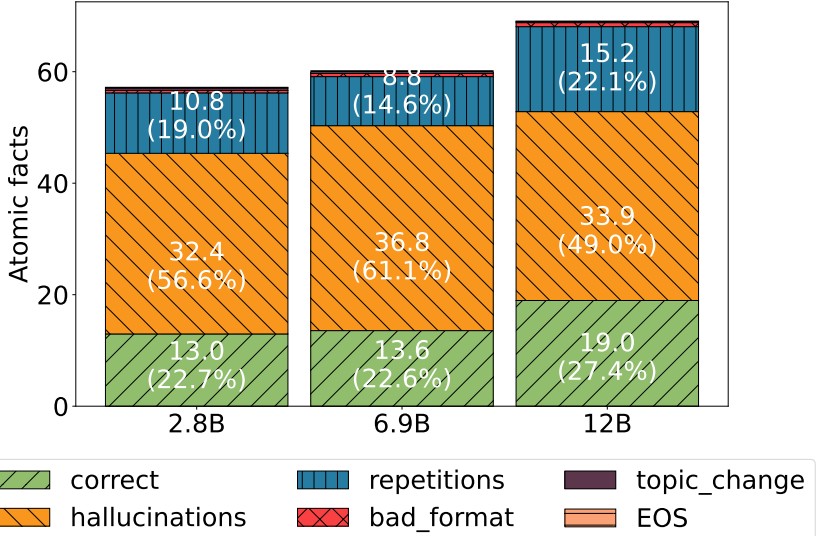

Figure 38: **Larger `Pythia` models hallucinate more when generating open-ended biographies on very popular entities.** When producing biographies of entities with very high frequency of appearance in Wikipedia, larger models generate more atomic facts, with an increasing rate of hallucinations.

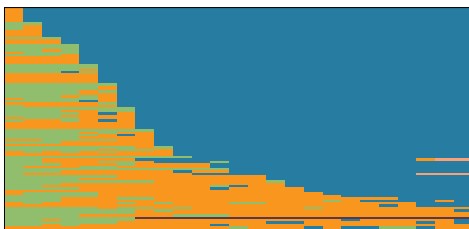

Figure 39: **Pyhtia-1.4B model shift fallback behavior during a single completion.** Order of fallbacks per generation on TRIVIAFACTS for a `Pythia`-1.4B model—each row represents a specific prompt with the 25 produced facts. Green marks correct answers, orange hallucinations and blue repeated facts. Purple sequences indicate a topic change. Questions are sorted by the number of consecutive repetitions.

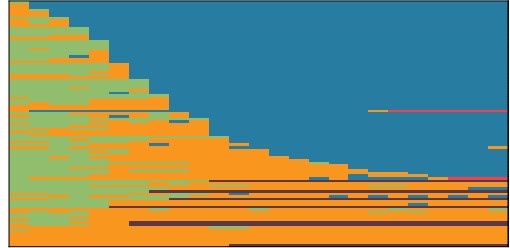

Figure 40: **Pyhtia-2.8B model shift fallback behavior during a single completion.** Order of fallbacks per generation on TRIVIAFACTS for a `Pythia`-2.8B model—each row represents a specific prompt with the 25 produced facts. Green marks correct answers, orange hallucinations and blue repeated facts. Purple sequences indicate a topic change, and red ones indicate a divergence to bad-format. Questions are sorted by the number of consecutive repetitions.

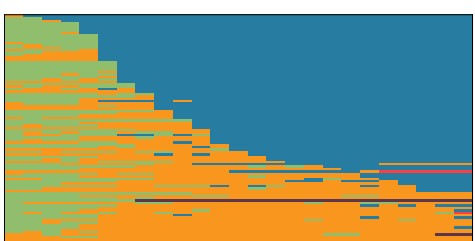

Figure 41: **Pyhtia-6.9B model shift fallback behavior during a single completion.** Order of fallbacks per generation on TRIVIAFACTS for a Pythia-6.9B model—each row represents a specific prompt with the 25 produced facts. Green marks correct answers, orange hallucinations and blue repeated facts. Purple sequences indicate a topic change, and red ones indicate a divergence to bad-format. Questions are sorted by the number of consecutive repetitions.

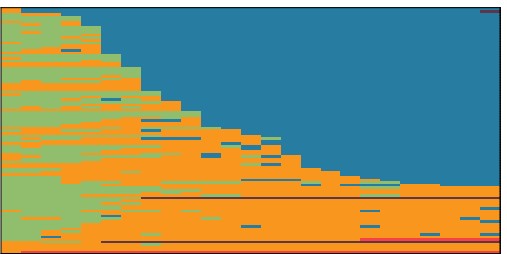

Figure 42: **Pyhtia-12B model shift fallback behavior during a single completion.** Order of fallbacks per generation on TRIVIAFACTS for a Pythia-12B model—each row represents a specific prompt with the 25 produced facts. Green marks correct answers, orange hallucinations and blue repeated facts. Purple sequences indicate a topic change, and red ones indicate a divergence to bad-format. Questions are sorted by the number of consecutive repetitions.

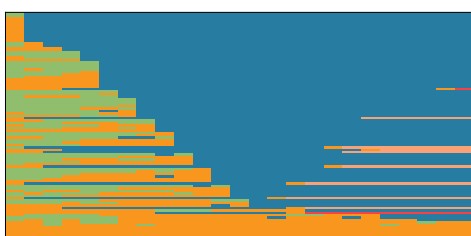

Figure 43: **OLMo-1B model shift fallback behavior during a single completion.** Order of fallbacks per generation on TRIVIAFACTS for a OLMo-1B model—each row represents a specific prompt with the 25 produced facts. Green marks correct answers, orange hallucinations and blue repeated facts. Purple sequences indicate a topic change, and red ones indicate a divergence to bad-format. Questions are sorted by the number of consecutive repetitions.

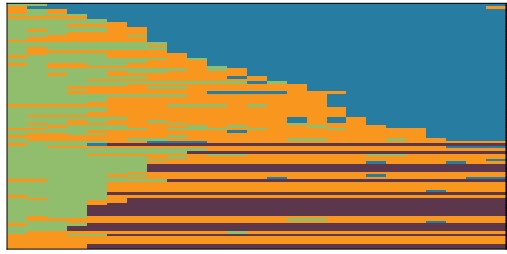

Figure 44: **OLMo-7B model shift fallback behavior during a single completion.** Order of fallbacks per generation on TRIVIAFACTS for a OLMo-7B model—each row represents a specific prompt with the 25 produced facts. Green marks correct answers, orange hallucinations and blue repeated facts. Purple sequences indicate a topic change, and red ones indicate a divergence to bad-format. Questions are sorted by the number of consecutive repetitions.

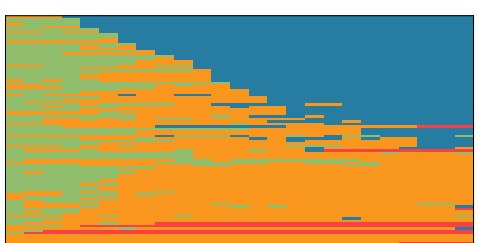

Figure 45: **Llama 2 7B model shift fallback behavior during a single completion.** Order of fallbacks per generation on TRIVIAFACTS for a Llama 2 7B model—each row represents a specific prompt with the 25 produced facts. Green marks correct answers, orange hallucinations and blue repeated facts. Purple sequences indicate a topic change, and red ones indicate a divergence to bad-format. Questions are sorted by the number of consecutive repetitions.

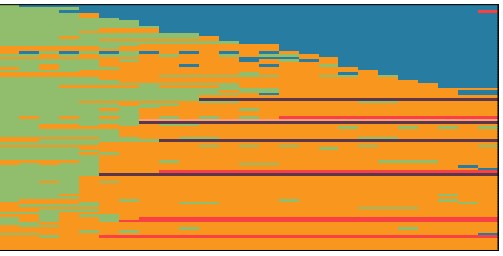

Figure 46: **Llama 2 70B model shift fallback behavior during a single completion.** Order of fallbacks per generation on TRIVIAFACTS for a Llama 2 70B model—each row represents a specific prompt with the 25 produced facts. Green marks correct answers, orange hallucinations and blue repeated facts. Purple sequences indicate a topic change, and red ones indicate a divergence to bad-format. Questions are sorted by the number of consecutive repetitions.

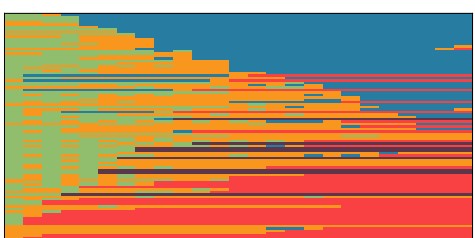

Figure 47: **Llama 3 8B model shift fallback behavior during a single completion.** Order of fallbacks per generation on TRIVIAFACTS for a Llama 3 8B model—each row represents a specific prompt with the 25 produced facts. Green marks correct answers, orange hallucinations and blue repeated facts. Purple sequences indicate a topic change, and red ones indicate a divergence to bad-format. Questions are sorted by the number of consecutive repetitions.

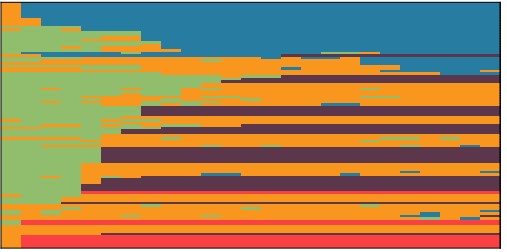

Figure 48: **Llama 3 70B model shift fallback behavior during a single completion.** Order of fallbacks per generation on TRIVIAFACTS for a Llama 3 70B model—each row represents a specific prompt with the 25 produced facts. Green marks correct answers, orange hallucinations and blue repeated facts. Purple sequences indicate a topic change, and red ones indicate a divergence to bad-format. Questions are sorted by the number of consecutive repetitions.

