# OpenReview forum: "From Loops to Oops: Fallback Behaviors of Language Models Under Uncertainty"
_ICLR.cc/2025/Conference — Submitted to ICLR 2025_

### Official Review · Reviewer_Rirk · 2024-11-01

**Soundness:** 3
**Presentation:** 4
**Contribution:** 3
**Rating:** 8
**Confidence:** 4

**Summary:**

The paper studies different fallback behaviors of LLMs that occur under uncertainty,
For this, the authors construct settings where the model is expected to be uncertain.
They evaluate various models differing in both size and how long they are trained and compare greedy vs. sampling-based decoding methods.
The authors find that "stronger" models (trained longer, more parameters) exhibit more complex fallback behaviors which means that more hallucinations are produced.

**Strengths:**

- The idea of "epistemic uncertainty by construction" is an interesting way of circumventing problems with quantification of uncertainty.

- Overall, the paper seems methodologically sound.

- The paper is very well-written and easy to follow.

- The ideas presented in the paper can help guide further works on combating (unwanted) fallback behaviors of LLMs

**Weaknesses:**

- Epistemic uncertainty as a whole is a quite rich concept that might warrant a larger discussion. In particular, the authors focus mainly on uncertainty about facts but mostly neglect other forms / sources of uncertainty that are also often seen as contributing to epistemic uncertainty. An example is L417-419: verbatim recollection is hypothesized to be more present in OOD settings than factual uncertainty but OOD settings are often also thought to give rise to epistemic uncertainty, because not enough training data is observed in the domain, leading to a lack of knowledge. For example, the authors could discuss the relation of epistemic uncertainty and aleatoric uncertainty [1] and the sources of epistemic uncertainty further [2]

- The main dataset is comparably small with only 95 items but also highly curated. Similarly, the other datasets are also fairly small, usually around 100 examples

- Some statements are a bit imprecise, for example: what is meant with internal capability to avoid hallucinations (L303) and what backs it up experimentally?

**Questions:**

Below are some questions and comments:

- Footnote 2 is a bit hard to understand without having read later parts of the paper

- it's not immediately clear to me why Datasets 1-3 have different levels of epistemic uncertainty, could you clarify this?

References

[1] Hüllermeier, Eyke, and Willem Waegeman. "Aleatoric and epistemic uncertainty in machine learning: An introduction to concepts and methods." Machine learning 110.3 (2021): 457-506.

[2] Baan, Joris, et al. "Uncertainty in natural language generation: From theory to applications." arXiv preprint arXiv:2307.15703 (2023).

---

> ### Author Response · Authors · 2024-11-17
>
> We thank the reviewer for their positive review and insightful comments. Below, we address the points raised and outline the improvements we will make following the reviewer’s suggestions.
>
>
> We agree that epistemic uncertainty warrants a broader discussion. We acknowledge that our initial focus on factual uncertainty underrepresents the broader scope of epistemic uncertainty as well as other types of uncertainty such as aleatoric. Following your suggestion, we have added to the paper a discussion about the topic, which we add here below.
>
> ```
> In this work, we focus on epistemic uncertainty that reflects a lack of knowledge, regardless of its root cause, such as training data limitations, model architecture, or optimization strategies. This approach aligns with the definitions provided by Hüllermeier and Waegeman (2020) ￼, where epistemic uncertainty is characterized as reducible through additional data or better modeling, and complements the broader uncertainty taxonomy discussed by Baan et al. (2023)￼. Additionally, Abbasi Yadkori et al. (2024) emphasize the importance of distinguishing epistemic uncertainty from aleatoric uncertainty, which arises from inherent randomness, particularly in complex outputs or out-of-distribution settings ￼. While our study primarily investigates epistemic uncertainty that covers knowledge-intensive use-cases of LLMs, we recognize the value of exploring aleatoric uncertainty and its implications for model behavior in scenarios involving ambiguous or out-of-distribution inputs, and leave such exploration for future work.
> ```
>
> We welcome further discourse about the topic, and would value the reviewer's  inputs to improve the above discussion.
>
>
> Our datasets are small but highly curated. The variance in results on individual questions is low, which ensures that when aggregated, the findings are statistically significant. If the reviewer finds it helpful, we are happy to include these variance estimations for the different experiments in the final revision.
>
> Clarifications and Improvements:
>
> - Footnote 2: We agree this may be unclear in the context it appears in. We will rephrase it as: “While the model may experience internal uncertainty when predicting the next token, this uncertainty might not be reflected in the logits. Once a fallback behavior is selected, it may elevate certain tokens, leading to high logit values.” This will be integrated more smoothly in the revised version.
> - Datasets with Different Levels of Uncertainty: We did not claim that the datasets represent distinct levels of epistemic uncertainty but instead tested various settings where uncertainty is present, such as varying popularity levels correlated with model accuracy. We will clarify this in the paper and welcome suggestions if the reviewer has additional concerns.
> - The “internal capability to avoid hallucinations” (L303) refers to the expectation models may be aware of their internal knowledge, and be able to refrain from producing hallucinations if asked to [1]. eWe will refine this phrasing and back it with clearer evidence in the revised version.
>
> We are grateful for the reviewer’s thoughtful suggestions, which already improved our paper. Thank you again for your constructive feedback!
>
> [1] https://arxiv.org/abs/2305.18153

---

> > ### Comment · Reviewer_Rirk · 2024-11-27
> > **Thank you for your response!**
> >
> > Dear Authors,
> >
> > thank you for your response and adding a further discussion about epistemic uncertainty, this addresses my questions for now!

---

### Official Review · Reviewer_PL8x · 2024-11-03

**Soundness:** 2
**Presentation:** 3
**Contribution:** 2
**Rating:** 5
**Confidence:** 4

**Summary:**

This paper mainly analyzes the fallback behaviors of large language models (LLMs), which include sequence repetitions, degenerate text, and hallucinations. This paper conduct various experiments to explore the fallback behaviors in models from the same family that differ by  the amount of pretraining tokens, parameter count, or the inclusion of instruction-following training. It further discuss factors influencing the fallback behaviors of a LLM, such as sampling methods, simplifying the question and prompt engineering. The experiments reveal the following phenomenon:
1. Increasing model strength through extra pretraining, more parameters, or instruction tuning shifts fallback behaviors from simple to complex, i.e., from repetitions to hallucinations.
2. While LLMs have some internal capability to avoid hallucinations, this fallback behavior is inherent to their generation scheme and is likely unavoidable with current decoding methods.
3. As models generate longer texts, they shift in their fallback behavior, first generating hallucinations and eventually producing degenerate text.

**Strengths:**

•	This paper provides a controlled analysis of LLM undesired behaviors, showing that strengthening models increases hallucinations over degenerate text and repetitions.
•	This paper demonstrates that during generation, LLMs undergo a phase shift from correct answers to hallucinations and then repetitions.
•	This paper indicates that methods to reduce text degeneration, like sampling, increase errors and hallucinations.

**Weaknesses:**

•	This paper seems to overclaim its contribution and conclusion. According to the datasets this paper used, the LLMs need to recall multiple facts that are related to the same topic. The other QA circumstances are ignored, including the simple QA like Natural Questions, multi-hop question answering like HotpotQA, open-ended long-formed question answering but beyond biography like ELI5. Also many other knowledge intensive tasks are also not discussed, which is also related with the epistemic uncertainty.
•	According to the paper, the fallback behaviors include sequence repetitions, degenerate text, and hallucinations. But degeneration is not discussed in the metric and experiment results.
•	The experiment setting of FQAMP seems not reasonable. If forcing uncertainty is essential, it is more natural to replace the subjects with a low popularity names. If replacing with made-up names and requiring LLMs to abstain, it is an OOD problem but not an uncertainty problem.

**Questions:**

•	If this paper focuses on “epistemic uncertainty” and do not explore other types of uncertainty, why this paper is titles “under epistemic uncertainty”.
•	Is the TRIVIAFACTS dataset evaluated? The question mentioned in the paper, which is “The following 25 colors are in the Olympic rings.” seems to mislead LLMs to repeat and hallucinate.

---

> ### Author Response · Authors · 2024-11-17
>
> We thank the reviewer for their detailed feedback. We believe the reviewer may have misunderstood some key points in our work, which we clarify below.
>
> QA Settings: Please note that our focus is on fallback behaviors under epistemic uncertainty rather than general QA performance. The suggested datasets, like NaturalQuestions and HotpotQA, are general QA datasets that focus on measuring factuality and reading comprehension when producing a single answer. Models today perform relatively well on these, so we are not sure why the reviewer suggests they are suitable for evaluating epistemic uncertainty. As noted in the paper, larger models’ factuality improvements are well established, and adding these datasets does not provide additional insight into the fallback behaviors of models when they do **not** have the right answer (as each question gets a single answer which is either correct or an hallucination).
>
> Knowledge-Intensive Tasks: We believe our evaluations already address uncertainty in knowledge-intensive tasks, particularly in Bio and Qampari, which require models to recall multiple facts about uncommon subjects and entities. If the reviewer has specific suggestions, we would be happy to consider them.
>
> Degeneration Metrics: Degeneration is explicitly addressed in Section 6.2 (lines 421–444) and referenced earlier in the paper (lines 209–210). To further highlight its importance, we have added an explicit discussion of the topic in the introduction, which will be reflected in the final version, ensuring readers can easily locate and engage with this analysis.
>
> Experimental settings: We stand by the FQAMP design, as introducing fabricated subjects reflects real-world queries with false premises [1],[2]. The entities in the Qampari dataset have varying levels of popularity, and replacing them with less-popular real entities could result in a less controllable setting. We emphasize that FQAMP complements other approaches, such as low-popularity entities, which we analyze in the Bio dataset. There, we explore the emergence of fallback behaviors across entities with increasing levels of popularity (lines 144–146).
>
> TriviaFacts Dataset: Perhaps there is some misunderstanding here. It is not designed to mislead models but to induce uncertainty by requiring responses with incomplete or uncertain knowledge. This naturally increases fallback behaviors, aligning with our research goals. We validated the questions with extensive manual annotation to ensure they serve this purpose and will clarify this in the paper.
>
> We hope these clarifications address the reviewer’s concerns and demonstrate the robustness of our approach. We would appreciate it if the reviewer reconsiders their scores, and are happy to engage in discussion if there are any further questions or concerns. Thank you for the constructive feedback.
>
> [1]  https://www.arxiv.org/abs/2407.12043
>
> [2] https://aclanthology.org/2024.emnlp-main.155/

---

> > ### Author Response · Authors · 2024-11-29
> >
> > We thank the reviewer once again and would like to draw their attention to the revisions we uploaded a few days ago, summarized in the general message.
> >
> > In response to the reviewer’s suggestions, we have added further justifications and discussion about the FQAMP dataset and made the discussion on text degeneration more explicit, with clear references in the introduction. Additionally, we have expanded the discussion regarding the inference setup for different models and ensured all implementation details are clearly provided to address concerns about the results with the Llama models.
> >
> > Moreover, as suggested by the reviewer, we conducted new experiments with top-p sampling, alongside additional models for both top-p and temperature sampling. These results are now included in Section 5.1, offering more evidence to support our conclusions.
> >
> > We would greatly appreciate it if the reviewer could review these updates, which we believe address their concerns, and reconsider their evaluation.
> >
> > Thank you,
> > Authors

---

### Official Review · Reviewer_tXwN · 2024-11-04

**Soundness:** 3
**Presentation:** 2
**Contribution:** 3
**Rating:** 5
**Confidence:** 4

**Summary:**

The paper studies how LLMs behave in the face of epistemic uncertainty, claiming that the LLMs “fall back” to various strategies, such as hallucinations, degenerate text, and more when this occurs. The study explores these fallback strategies across several model parameters, including model size, pretraining tokens, instruction tuning, and decoding algorithms.

The paper presents some interesting findings such as:

1. Larger models tend to be more factual but are more likely to hallucinate as a fallback strategy rather than generate repetitive or poorly formatted text.
2. As the number of pretraining tokens increases, the number of factual responses also increases, though the model may hallucinate more frequently.
3. Instruction tuning makes the model's fallback strategies more complex without significantly changing the percentage of factual answers.
4. Increasing the sampling temperature slightly reduces factuality but decreases repetitiveness and occasionally enhances creativity.


The authors conduct experiments on several different models, checkpoints, and datasets.

**Strengths:**

The paper analyzes fallback mechanisms when the model encounters uncertainty—an important and timely issue. The experimentation is extensive, with tests on multiple models across various datasets to validate its conclusions. The findings are interesting and hold significant value for other researchers working on LLM hallucinations. Some of the conclusions seem less surprising (although empirical verification of them is still useful)

While the paper has notable strengths, there are some concerns (see weaknesses). I am open to raising my score once these issues have been addressed.

**Weaknesses:**

1. **Presentation Improvements**

   - The paper’s presentation could be refined. In its current form, it may take considerable time for readers to extract the main takeaways.
   - There are multiple task settings, and results can sometimes be difficult to interpret due to the variety of datasets used for each claim. Standardizing these settings could improve clarity.
   - In some cases, important results are presented only in the appendix, which can be frustrating for readers. Additionally, the graph captions could be more informative (for instance, specifying the dataset used for Figures 3 and 4).
   - Terminology is sometimes inconsistent—for example, the term "stronger models" is used, but it’s unclear what this means.

2. **Experimentation Concerns**

   - Why are instruction-tuned models tested only on TriviaFacts? Although list-based questions may simplify evaluation, this limits conclusions to a single, small, self-created task (hence weakening the claim)
   - Some parameters are analyzed independently, even though they often co-occur. For example, model size could be varied specifically within instruction-tuned models. This approach sometimes leads to inconsistencies. In Figure 21, llama-3-8B-inst is shown to be more factual (% factual) and hallucinates significantly more than llama-3-70B-inst—contradicting the earlier finding that larger models tend to produce more factual answers while hallucinating more. This also appears inconsistent with broader findings, such as in the Llama-3 report, where llama-3-70B is generally noted as more accurate.

   These outliers and inconsistencies warrant further discussion.

**Questions:**

See weaknesses

---

> ### Author Response · Authors · 2024-11-17
>
> We thank the reviewer for their thoughtful comments and feedback, and we are pleased they found the analysis extensive and the findings valuable.
>
>
> Presentation:
>
> We appreciate the reviewer’s points about clarity. Other reviewers have explicitly highlighted our presentation, particularly the “takeaway” boxes in each section, as a strength in summarizing key results. While we believe these elements enhance readability and emphasize our main conclusions, we welcome specific suggestions during the rebuttal period. We have already refined graph captions (e.g., specifying datasets for Figures 3 and 4) and adjusted terminology such as “stronger models,” which will be reflected in the final version. Any additional comments to further improve the paper’s readability are welcome.
>
>
> Experimentation:
>
> Our results are robust and conclusive for the studied settings, supported by extensive evaluations across multiple models, datasets, and parameters. We are happy to follow the reviewer’s suggestion and expand results for instruction-tuned models on additional datasets to strengthen this further.
>
> Regarding the Llama models, we clarify that we used Llama 3.1, whereas the cited report references Llama 3.2. Additionally, for the 70B model, we used FP8 quantization, which can slightly affect its performance. Finally, as noted in the appendix, some of the released open weights that were released have been found to behave unexpectedly. For completeness, we will include results with Llama 3.2 in the final version, as per the reviewer's request.
>
>
> We believe our results provide a strong foundation for studying fallback behaviors in LLMs and are conclusive for the scenarios examined. We hope these clarifications resolve the reviewer’s concerns and hope they will reconsider their assessment of our work. We are happy also to engage in discussion and answer any follow up questions the reviewer may have.

---

> > ### Comment · Reviewer_tXwN · 2024-11-22
> > **Response**
> >
> > Thanks for your response!
> >
> > I appreciate the inclusion of takeaway boxes, as they effectively highlight the main findings of the experiments. However, I found it challenging to dig deeper into the experimental details in several cases. I'm glad to see that the authors are actively working to address this issue.
> >
> > > Additionally, for the 70B model, we used FP8 quantization, which can slightly affect its performance
> >
> > **This detail was not mentioned in the paper. Comparing a quantized model to a full-precision model can introduce inconsistencies and this should be mentioned**
> >
> > Currently, the paper (and appendix) feels overloaded with experiments, making it difficult to follow the experimental settings and datasets used across the various sections.
> > As I mentioned there are inconsistencies in the datasets used to compare across different claims which makes following the storyline even more difficult.
> > Despite this, the paper provides valuable insights into LLM hallucination and how the model behaves when uncertain. I strongly believe a round of revisions could significantly enhance its clarity and impact.

---

> > > ### Author Response · Authors · 2024-11-29
> > >
> > > We thank the reviewer once again for their valuable feedback for suggesting that our work “ provides valuable insights into LLM hallucination and how the model behaves when uncertain”.  We would like to bring their attention to the revisions we uploaded a few days ago, summarized in the general message.
> > >
> > > Following the reviewer’s suggestions, we have conducted additional experiments, which are now incorporated into the revised submission. We have also clarified and improved the presentation of the setup, enhanced captions, and ensured consistent terminology, as recommended by the reviewer. Additionally, while the appendix contains many results reflecting the breadth of our experiments, we have ensured that the main results are included in the main text, with a clear structure to improve readability.
> > >
> > > We have also added further discussion regarding the setup used for inference with different models and made sure all implementation details are provided clearly. Moreover, we conducted and analyzed many new experiments with instruction-tuned models to further investigate the impact of this method on enhancing model capabilities and the emergence of fallback behaviors, and discuss the inconsistencies as noted by the reviewer.
> > >
> > > We deeply appreciate the reviewer’s thoughtful and constructive suggestions, which have led to significant improvements in our work. We would be grateful if the reviewer could review these changes, which we believe address their concerns, and reconsider their evaluation. We will also be happy to address any remaining clarifications before the end of the discussion period.
> > >
> > > Thank you,
> > > Authors

---

### Official Review · Reviewer_CG5v · 2024-11-06

**Soundness:** 3
**Presentation:** 4
**Contribution:** 2
**Rating:** 5
**Confidence:** 4

**Summary:**

This paper looks at “fallback” behaviors in large language models in the face of epistemic uncertainty, specifically focusing on repetitive text, degenerate phrasing, and hallucinations. They set up scenarios to control the presence of epistemic uncertainty and characterise the outputs of various LLMs, evaluating how the number of parameters, amount of pretraining data, instruction-following training, and decoding impact the type of fallback behavior exhibited. They observe what they call a “fallback hierarchy” where as models become more uncertain (or if they are weaker to begin with), they rely more on basic degenerate behaviors. They posit that fallback behaviors are interconnected, based on the observation that models transition from one behavior to another as uncertainty decreases. Importantly, they find that larger, more sophisticated models also exhibit fallback behaviors in the face of uncertainty, even if they may be more sophisticated fallback behaviors and that some techniques claimed to mitigate degenerate behaviors have other adverse effects or only a limited ability to help

**Strengths:**

* The work offers a thorough analysis across various LLM families of different types of frequently-observed degenerate behavior
* The experimental setup seems like a reasonable way to control for epistemic uncertainty
* There are interesting empirical findings about the prevalence and characteristics of degenerate outputs from modern LLMs

**Weaknesses:**

* The contributions of the work are limited:
    * empirical observations are made but no insights about what the source of these behaviors are or how they can be fixed are given
    * The analysis is limited to the presence of epistemic uncertainty, but there are other scenarios in which degenerate behaviors occur (e.g. when adversarial prompts are given or in the face of aleatoric uncertainty) so we’re still far from a comprehensive empirical analysis of degenerate behaviors
* The setups are rather contrived and since the work's results are only empirical, its not clear what the practical implications of them are
* While the authors claim to explore the impact of decoding on these fallback behaviors, they seem to only look at greedy decoding and temperature sampling. They also seem to only explore one prompt as a “mitigation” technique
* Some claims don’t feel adequately supported. E.g., “fallback behaviors… emerge…unavoidably under uncertainty” Even though a comprehensive set of prevention techniques weren’t explored

**Questions:**

I don’t understand what is meant in the sentence “Moreover, introducing randomness reduces the number of correct facts, which could be attributed to the random skipping of correct facts that have low confidence in the model.” Could you clarify?

---

> ### Author Response · Authors · 2024-11-17
>
> We thank the reviewer for their thoughtful feedback.
>
> W1: Our focus is on characterizing LLM behavior under epistemic uncertainty, an important scientific question that complements existing works on mechanisms behind hallucinations or repetitions. We believe that, given how central these behaviors are, such analysis work is valuable in itself and often inspires practical research [1]. Understanding the sources of such behaviors is a substantial challenge in itself [2] and outside the scope of this work.
>
> W2: Expanding our evaluation to other uncertainties, like aleatoric uncertainty or adversarial prompts, would require adapting our “uncertainty by construction” framework, which is non-trivial but valuable future work. Practical implications of our findings do exist; for instance, scalable oversight [3] highlights how larger models’ fallback behaviors become harder to detect, which is an important issue for practitioners. Our work connects these behaviors and so future work may be able to intervene in the model to fallback into repetitions instead of hallucinations, or to develop better detectors for uncertainty by dissecting the similarities between these behaviors.
>
> W3: Regarding decoding and mitigation, we focused on common techniques (greedy and temperature sampling) and acknowledge that exploring additional strategies such as top-p sampling is interesting, and we will be happy to include such analysis in the final version.
>
> W4: Our observations across an extensive suite of models, datasets, and settings show consistent trends regarding the emergence and ordering between the studied behaviors. While our experiments largely focused on natural settings, where the system/user does not explicitly attempt to prevent certain behaviors, to study the model's natural behavior, we have explored various techniques to prevent models from producing hallucinations: using in-context examples (Section 5.3), asking models to list only the facts they know about an entity rather than forcing them to recall a predetermined number of facts (Section 5.2), providing explicit instructions for instruction-tuned models (lines 360–369), and employing random sampling (Section 5.1). As mentioned above, we will also add experiments using top-p sampling.  We believe these experiments provide sufficient evidence to support our claims. If the reviewer has concrete suggestions for additional mitigation techniques or alternative phrasings that would be more suitable, we are happy to consider them.
>
> Q1: By “introducing randomness reduces the number of correct facts,” we mean that probabilistic sampling can lead to missing correct facts. For example, if the first token for a correct fact has a probability of 0.51, it would be selected in greedy decoding but may be skipped with random sampling.
>
> We hope this clarifies our contributions and inspires further dialogue.
>
> [1] https://aclanthology.org/2024.emnlp-main.181/
>
> [2] https://arxiv.org/abs/2202.07646  (ICLR 2023)
>
> [3] https://arxiv.org/abs/2211.03540

---

> > ### Comment · Reviewer_CG5v · 2024-11-22
> > **Response**
> >
> > Thank you for the clarifications. I still feel like the contribution of the work is limited though (the main reason for the score I gave) and will thus keep my score the same. Also, continuing on the point from reviewer Rirk, I think a more explicit discussion about the variance/statistical significance of results would be very helpful, given the small size of the curated dataset

---

> > > ### Author Response · Authors · 2024-11-29
> > >
> > > We thank the reviewer once again and would like to draw their attention to the revisions we uploaded a few days ago, summarized in the general message.
> > >
> > > In response to the reviewer’s suggestions, we have conducted additional experiments, now included in the revised submission. Specifically, addressing the reviewer’s questions, we provide further evidence on the impact of randomness on factual answers, including new results with additional decoding methods which further strengthen our findings, as well as an expanded discussion and clarification of the practical implications of our work.
> > >
> > > We would greatly appreciate it if the reviewer could review these updates, which we believe address their concerns, and reconsider their evaluation.
> > >
> > > Thank you,
> > > Authors

---

### Author Response · Authors · 2024-11-25
**Revision updates and general response**

We sincerely thank all reviewers for their thoughtful and constructive feedback, which has helped us significantly improve the clarity and quality of the paper. Below, we address the changes made in response to the reviewers’ suggestions, indicating where these revisions are reflected in the updated manuscript.

1. Clarifications on Key Points:
   - In response to Reviewer Rirk’s comment about footnote 2 being difficult to understand, we revised its phrasing for clarity. It now appears as footnote 3 in the new version.
   - Reviewer CG5v raised a question about the impact of randomness on factual answers. We have added footnote 7 to clarify why randomness reduces the correctness of answers.
2. Discussion Enhancements:
   - As suggested by Reviewer Rirk, we expanded our discussion of other types of uncertainties (e.g., aleatoric uncertainty) and their relationship to epistemic uncertainty. This is now included in Section 7 (related work, _Uncertainty in language modeling_, lines 503-519), along with citations to relevant literature, including Hüllermeier et al. (2021) and Baan et al. (2023).
   - Following Reviewer CG5v’s comments about the practical implications of our work, we added a discussion of these points to Section 8 (Conclusion and Discussion, lines 532-539).
3. Dataset and Experimental Justifications:
   - Addressing Reviewer PL8x’s concerns regarding FQAMP, we added a justification for its importance and its alignment with natural scenarios in Section 3, where the dataset is defined.
   - Reviewer tXwN noted inconsistencies in terminology such as “stronger models.” We have now provided a formal definition of this term in footnote 2 in the introduction, ensuring consistency throughout the paper.
4. Improved Presentation:
   - Reviewer tXwN suggested improving the clarity of figures by specifying the dataset used in the captions. We have updated captions for Figures 2, 3, 4, 5, and 7 accordingly.
   - To make the paper easier to follow, we verified that all significant results are now presented in the main body, even if full figures are still in the appendix. We thank Reviewer tXwN for highlighting this, and we agree that the revised structure is more reader-friendly.
   - To address Reviewer PL8x’s feedback on degeneration metrics, we made the discussion of degeneration explicit from the beginning of Section 6.2.
5. Additional Experiments:
   - Following Reviewer tXwN’s request, we conducted additional experiments with instruction-tuned models across more datasets. These results have been added to Sections 4.3 and 5.3, further strengthening our findings.
   - As requested by Reviewer PL8x, we performed new experiments with top-p sampling, alongside additional models for both top-p and temperature sampling. These results are included in Section 5.1, providing more robust evidence for our conclusions.
6. Addressing LLaMA Concerns: Reviewer PL8x and Reviewer tXwN raised concerns about inconsistencies in LLaMA results. We added detailed explanations of how models were loaded and potential issues with some LLaMA checkpoints weights. These are now included in Section 3 and footnote 5, with further discussion in Appendix E.1.
7. Structural Improvements: Reviewer tXwN suggested adding an outline of the paper to the introduction. We have now included a paragraph at the end of the introduction that outlines the structure of the paper and explicitly mentions where key topics, such as degenerate text, are discussed (e.g., Section 6).

We deeply appreciate the reviewers’ efforts to engage with our work and provide actionable suggestions. We believe these changes have strengthened the paper and made it more accessible and informative. As we have addressed the reviewers’ concerns and conducted the experiments they requested, we would sincerely appreciate it if they could reconsider their evaluation and consider increasing their scores. Thank you for your time and valuable input.

---

### Author Response · Authors · 2024-12-03

We would like to once again thank the reviewers for their comments and suggestions. We have made considerable efforts to address the concerns raised, including running additional experiments and revising the paper accordingly. As the discussion period is nearing its conclusion and we have not received further comments, we would highly appreciate it if the reviewers can take our responses into account and reevaluate their assessment of the paper.
Thank you again for your time and constructive feedback.

---

### Meta-Review · Area_Chair_Y6sB · 2024-12-19

**Metareview:**

The paper investigates fallback behaviours in large language models (LLMs) under epistemic uncertainty, categorizing them into sequence repetitions, degenerate text, and hallucinations, and proposes a "fallback hierarchy" that connects these behaviours. Experiments are performed across various model configurations, including variations in pretraining data, parameter sizes, and decoding methods, demonstrating consistent patterns in how LLMs exhibit fallback behaviours as uncertainty increases.

Despite the relevance of the topic and the thoroughness of some experiments, the paper has several shortcomings. While the authors provide an interesting characterization of fallback behaviours, the claims about their inevitability and possible mitigation remain insufficiently supported. The experimental setups rely heavily on small, somewhat contrived datasets that limit generalizability, and key findings are weakened by inconsistent terminology, a lack of clarity in experimental descriptions, and an overreliance on appendices for important results. Although the authors conducted additional experiments and added discussions on related uncertainties during the rebuttal period, these revisions did not adequately address concerns about the paper’s scope, generality, and practical implications. Given the limited generality of the findings, the unresolved methodological and presentation issues, and the overstated contributions, I recommend rejecting the paper.

**Additional Comments On Reviewer Discussion:**

During the rebuttal period, the reviewers raised concerns about the limited scope, insufficient support for the paper's claims, and issues with clarity and experimental design. Reviewer CG5v highlighted that the contributions were largely empirical, lacked theoretical depth, and were limited in generality due to the narrow focus on epistemic uncertainty without addressing other forms like aleatoric uncertainty. Reviewer tXwN pointed out inconsistencies in terminology, difficulties in interpreting experimental setups, and concerns about the reliance on small, curated datasets, suggesting that the findings may not generalize. Reviewer PL8x critiqued the design of the FQAMP experiment, arguing that it reflected out-of-distribution behaviour rather than epistemic uncertainty and found the analysis of degeneration metrics inadequate. While Reviewer Rirk found the paper methodologically sound and commended its presentation, they noted that the exploration of epistemic uncertainty could be broadened to include other sources or types.

The authors responded by conducting additional experiments, such as including results with top-p sampling, revising graph captions for clarity, and expanding discussions on uncertainty frameworks and degeneration metrics. They argued that the small, curated datasets ensured statistical significance and provided explanations for certain inconsistencies in the LLaMA model results. Despite these efforts, the reviewers remained largely unconvinced. Reviewer CG5v maintained that the revisions did not sufficiently address the narrow scope, overstated contributions, and limitations in generalizability. tXwN appreciated the changes to clarity and additional experiments but found the revisions insufficient to resolve issues with presentation and experimental design.

In weighing these points, I considered the authors’ efforts to address the feedback but concluded that the fundamental concerns—such as the limited scope, reliance on small datasets, and lack of actionable insights—remained unresolved. While the topic and findings are relevant, the paper does not sufficiently advance understanding or provide practical contributions to justify acceptance. These considerations informed my decision to recommend rejecting the paper.

---

### Decision · Program_Chairs · 2025-01-22

Reject